# A statistical-parametric model of tropical cyclones for hazard assessment

William C. Arthur[1]

[1]Geoscience Australia, Canberra, ACT, 2601, Australia

*Correspondence to*: Craig Arthur (craig.arthur@ga.gov.au)

**Abstract.** We present the formulation of an open-source, statistical-parametric model of tropical cyclones (TCs) for use in hazard and risk assessment applications. The model derives statistical relations for TC behaviour (genesis rate and location, intensity, speed and direction of translation) from best-track datasets, then uses these relations to create a synthetic catalogue based on stochastic sampling, representing many thousands of years of activity. A parametric wind field, based on radial profiles and boundary layer models, is applied to each event in the catalogue that is then used to fit extreme value distributions for evaluation of return period wind speeds. We demonstrate the capability of the model to replicate observed behaviour of TCs, including coastal landfall rates which is of significant importance for risk assessments.

## 1 Introduction

Tropical cyclones (TCs) present a significant physical and economic threat to Australian communities. Around 20% of reported costs from natural disasters arise from TCs (Handmer et al., 2016), while over 30% of insured losses are caused by TCs (Chen, 2004), due to extreme winds and riverine and coastal storm surge flooding. Minimising the losses in the built environment from these events can be approached in a range of ways. In Australia, the Wind Loading Standard (AS/NZS 1170.2, 2011) specifies minimum design loads for buildings under the action of wind loading. Design loads vary across the country, depending on the sources and magnitude of winds, in an effort to minimise average annual losses across the country. Areas around the northern coastline have higher design loads, due to the exposure to TCs which generate higher wind speeds than mid-latitude storms. Design criteria are defined with reference to a likelihood of exceedance over the expected lifetime of residential structures – this is a 10% likelihood in 50 years, commonly described as a (approximately) 1-in-500 year average recurrence interval (ARI).

The historical record of TCs in the Australian region covers barely 100 years (Kuleshov et al., 2010). Of that record, only the last 30 years includes reasonably consistent information based on satellite data to assess the intensity of TCs. The short length of record makes it difficult to infer ARI wind speeds due to TCs at ARIs greater than 100 years (Emanuel and Jagger, 2010; Jagger and Elsner, 2006; Sanabria and Cechet, 2007). It is a common approach to use stochastic simulations to estimate the wind speeds to establish building design standards and for assessing TC risk (Vickery et al., 2009). Many of these models exploit the statistical characteristics of TC behaviour to generate catalogues of synthetic events (Emanuel et al., 2006; Hall and Jewson, 2007; James and Mason, 2005; Li and Hong, 2014; Nakajo et al., 2014; Powell et al., 2005; Rumpf et al., 2007).

In this vein, Geoscience Australia has developed a statistical-parametric model of TC behaviour — called the Tropical Cyclone Risk Model (TCRM) — to generate synthetic event sets that represent many thousands of years of TC activity. TCRM is designed to run on desktop computers with modest computational resources available, but is scalable to large multi-processor systems. As such, the model forgoes the more computationally intensive dynamical approach used in some TC hazard models (Emanuel et al., 2006). Instead, we use an autoregressive process to model synthetic TC tracks, including intensity, and use a two-dimensional parametric model to describe the TC wind field.

TCRM is unique in that it is freely available for use in hazard and risk assessment applications. There are a number of published stochastic models (noted above) however these models are as a general rule, not publically available (exceptions include the model of Powell et al., 2005) and at no cost. Further, the formulation of these models may preclude application in regions other than those demonstrated in publications. That is, they may be tailored to the region where they are applied. TCRM is formulated such that it is largely independent of the region being simulated, though some components are derived using regionally-specific data. Where there are region-specific formulations in a component of the model, these can be readily adapted for different regions. TCRM is also an open-source software package, enabling users to contribute to ongoing development of the model and influence the future directions of development priorities.

While the primary purpose for developing the model is to evaluate TC severe wind hazard, it can be configured to rapidly evaluate the swath of destructive winds from a single TC at high temporal and spatial resolution. In this configuration, a 2-dimensional wind field at 0.02° horizontal resolution, covering the entire track of a TC, is calculated in a matter of minutes. Using TCRM in this manner, we have evaluated the impact of individual TCs on Australian communities, with applications in emergency management and urban planning (Arthur et al., 2008; Krause and Arthur, 2018).

In Section 2, we describe the data used to develop and evaluate TCRM. Section 3 provides details of the track generation component of the model, and Section 4 describes the wind field modelling process. Section 5 describes the use of extreme value distributions to calculate ARI wind speeds, and Section 6 presents some initial results using TCRM in the Australian region.

## 2 Data

The analysis of TC wind hazard is based on historical observations of TC events and their characteristics. Specifically, the essential fields required for running the model include the date and time of TC observations, the location (longitude and latitude), intensity (central pressure) and a flag identifying unique TC events. Additional fields, such as the radius to maximum wind ($R_{max}$) and the pressure of the outermost closed isobar ($p_{oci}$) can be included, though are not essential.

For the Australian region, we use the International Best Track Archive for Climate Stewardship (IBTrACS: Knapp et al., 2010), which provides the most complete global set of historical TCs (Fig. 1). IBTrACS provides the date, time, position (longitude and latitude) and estimated central pressure of TCs in the southern hemisphere every 6 hours (or more frequently) for seasons between 1981 and 2016 (36 years). The data have been quality controlled and provide a homogeneous set of TC records from World Meteorological Organisation-sanctioned forecast agencies. This time period does exclude some historically significant storms, which is more so due to their impacts on the community rather than any physical characteristics. However, for testing and development, homogeneity of the input dataset is prioritised over the length of record.

Further, the absolute accuracy of the input data is viewed as a source of uncertainty in the hazard values presented here. For example, Courtney and Burton (2019) reported on progress to improve the best track archive in Australia, noting the reassessment of intensity due to improved reanalysis methods. Such changes in the intensity values will flow through the hazard model to produce changes in the likelihood of extreme wind speeds. A thorough treatment of the accuracies arising from changes in the best track is warranted (Harper et al., 2008), and the hazard values herein should be considered as only one view of the true wind hazard arising from TC events. Yet another aspect that remains to be explored is the effect of centennial and longer variability in TC activity (Haig et al., 2014; Nott et al., 2007).

There are some attributes of TCs that are not reported in the IBTrACS dataset. The radius to maximum wind ($R_{max}$) and pressure of the outermost closed isobar ($p_{oci}$) are two useful values that can provide additional constraints on the intensity and size of TCs. For these variables, we use data obtained from the Joint Typhoon Warning Center (2017), spanning

2002-2016 (15 years). This data is used to develop parametric models for these variables (described in Section 3), which are then used in the stochastic track generation process.

It is possible to use data sources other than observational best track archives as input to TCRM. For example, Siqueira et al. (2014) used tropical cyclone like vortices (TCLVs) extracted from global circulation models as a source of track data for evaluating TC wind hazard in the South West Pacific. After correcting the intensity distribution of the TCLV data, the resulting hazard assessment provided quantitative estimates of the projected change in TC wind hazard.

## 3 Model software

The TCRM software has been developed at Geoscience Australia as a free, open-source software package. It is written in Python (version 2.7), utilising the Numerical Python "NumPy" (van der Walt et al., 2011), Scientific Python "Scipy" (Jones et al., 2001), python-netcdf4 (Unidata, 2018), Pandas (McKinney, 2010), Matplotlib (Hunter, 2007) and Seaborn (Waskom et al., 2018) packages for statistical and visualisation functions. Additionally, we use some C code for optimisation. The software is available from Geoscience Australia's GitHub repository (Geoscience Australia, 2018), and users can contribute to further development of the model. For this study, we used TCRM version 2.1 (commit reference 8cd4c22: https://github.com/GeoscienceAustralia/tcrm/releases/tag/v2.1), and made use of the National Computational Infrastructure's High Performance Computing systems for executing the simulations and analysis of the results.

Simulation times are dependent on the extent of the domain, and the number of simulated years. For the domain used in this paper, the data processing and statistical analysis stages take around 15 minutes to complete on a modern desktop computer. The generation of tracks for a 10,000 year simulation takes around 5 to 6 CPU hours (2.6GHz clock speed), while the corresponding wind fields (a total of around 160,000 separate events for this simulation) take around 3,000 CPU hours. The determination of ARI wind speeds requires a similar amount of CPU time, but the majority is consumed in reading the required data from the wind field files.

## 4 Tropical cyclone track model

TCs in the Australian region often display complex behaviour, with many tracks exhibiting sudden turns (e.g. TC George, 2007) and loops (e.g. TC Hamish, 2009). Despite this behaviour, the translation speed and bearing of TCs still display significant autocorrelation (Fig. 2 and 3), while there is also a moderate autocorrelation in the rate of pressure change across the entire region (Fig. 4).

The track model is based on the approach used by Hall and Jewson (2007) and Rumpf et al. (2007), utilizing a lag-1 autoregressive technique to model the future behaviour of each synthetic TC. We extend this autoregressive technique and apply it to the intensity (minimum central pressure) of the simulated TCs as well as the track behaviour.

Users specify a simulation domain, over which the TC wind hazard will be evaluated (Fig. 1). To ensure the simulated events capture the complete range of potential tracks entering this domain, an expanded track domain is defined. The track domain is determined by examining the extent of all historical tracks that enter the simulation domain. The frequency and behaviour of simulated TCs is then determined on the basis of observed events in the track domain.

The model is trained on the observed track data, using a series of 1° by 1° grid cells across the track domain to capture spatial variability in the descriptive statistics (mean, standard deviation and lag-1 autocorrelation coefficient) for selected TC parameters (speed of forward motion, bearing, intensity and, where available, $R_{max}$). For each grid cell, a minimum of 100 valid observations are required before descriptive statistics are calculated. If there are insufficient valid observations, then the search area is expanded in steps of 1° zonally (east-west) and 0.5° meridionally (north-south) – to the maximum

extent of the track domain – until sufficient observations are found. Distributions for at-sea and over-land conditions are calculated separately to allow for different behaviours in these circumstances – specifically intensity in near-coastal areas. Regression models are used to control specific sub-components of the track model – $R_{max}$, $p_{oci}$ and landfall decay rate. These regression models are derived from observed data in the Australian region, but could equally be adapted to other regions. The code repository provides access to the analysis tools used to determine these regression models, and can be used to re-evaluate the regressions for other basins. The model is intended to be applied to regional basins, rather than a global domain, but the ability to adapt these regression models allows users to run in basins other than Australia.

## 4.1 Genesis

Genesis of TCs is modelled as a Poisson process based on historic frequency in the track domain, with locations of genesis randomly sampled from a 2-dimensional probability density function (PDF) of historic genesis points (Fig. 5). The PDF is generated using multivariate kernel density estimation (Silverman, 1986), utilizing a 2-dimensional Gaussian kernel. The PDF for genesis at a location $(\lambda, \phi)$ is:

$$f(\lambda, \phi) = \frac{1}{2\,\pi N |L|^2} \sum_{i=1}^{N} \exp\left(-\frac{d_i^2}{2L^2 L}\right),$$

(1)

where $N$ is the number of genesis points, $d_i$ is the distance between genesis point $i$

$\varepsilon$ is a random value sampled from a logistic distribution with zero mean and unit variance. A logistic distribution is used because the heavier tails provide a better representation of the distribution of residuals. Further, comparisons of full track simulations gave qualitatively better results when using the logistic distribution. A corresponding approach is used for the bearing (direction of movement) of simulated TCs.

Intensity, measured as the minimum central pressure $p(t)$, is also modelled in a similar manner, except it is the rate of change of intensity $\dot{p}(t)$ that is predicted at each time step $t$, rather than the intensity itself as described by Eqs. (5-7):

$$p(t) = p(t-1) + \dot{p}(t)\Delta t , \tag{5}$$

$$\dot{p}(t) = \mu_{\dot{p}}^i + \sigma_{\dot{p}}^i \chi^i(t) , \tag{6}$$

$$\chi^i(t) = \alpha_{\dot{p}}^i \chi^i(t-1) + \phi_{\dot{p}}^i \varepsilon , \tag{7}$$

where $\Delta t$ is the model time step in hours. The statistics for central pressure rates of change ($\mu_{\dot{p}}^i$ and $\sigma_{\dot{p}}^i$) are normalised to be in units of hPa hour$^{-1}$. $\alpha_{\dot{p}}^i$ and $\phi_{\dot{p}}^i$ are dimensionless and have the corresponding definition to that for $\alpha_v{}^i$ and $\phi_v{}^i$. The maximum achievable central pressure of a TC is set to $\mu_p^i - 5\sigma_p^i$, and is a purely statistical bound. However, we note that potential intensity (Holland, 1997) is potentially a more instructive limit, and we are presently working on enhancements that will consider this.

$\dot{p}$ is preferred to absolute pressure deficit due to the lower lag-1 autocorrelation in the tendency values, making it more akin to a true Markov process than simulating absolute pressure deficit. Figure 7 shows the autocorrelation for both $\dot{p}$ and $p$ for a selected grid cell in the Coral Sea. In this case, the lag-1 autocorrelation of $\dot{p}$ is 0.3, compared to that of $p$ which is 0.79. Using absolute values leads to rapid and almost one-way variation (i.e. constant increase or decrease) in the intensity. There remains a strong autocorrelation beyond lag-1 for absolute pressure values, but not for pressure tendency values (Fig. 8). Figure 9 shows the time history of central pressure of a small sample of tracks that are generated from a single genesis point (155°E, 20°S) and the same initial central pressure (995 hPa). One storm weakens rapidly over the first 12 hours. The remaining storms take between 30 and 200 hours to attain maximum lifetime intensity.

Initial values for $\dot{p}$, $v$ and storm bearing $\theta$ are sampled from the observed distributions of initial values in the initial grid cell, based on the randomly selected genesis point.

## 4.3 Radius to maximum winds

Where sufficient observed data is available, $R_{max}$ is modelled in a similar manner to intensity – i.e. an autoregressive model of the rate of change in $R_{max}$, with statistics calculated from the observed values. In the southern hemisphere, $R_{max}$ has only been recorded consistently since 2002 by the JTWC (Fig. 10). This means there is generally insufficient data to develop the autoregressive model with confidence across the entire model domain.

In the case of insufficient observations, a parametric model of $R_{max}$ is used, based on the model of Powell et al. (2005), and derived using recorded $R_{max}$ values from 2002-2016 JTWC records for the South Pacific and South Indian Ocean basins (n=3033):

$$\ln R_{max} = 3.543 - 0.00378\Delta p + 0.813 \exp(-0.0022\Delta p^2) + 0.00157\lambda^2 + \varepsilon , \tag{8}$$

where $\Delta p$ is the central pressure deficit (hPa), $\lambda$ is the latitude (degrees) and $\varepsilon$ is a random normal variate with mean $\mu = 0$ and variance $= 0.335$, which is held constant for the life of each individual simulated TC. The functional form is selected to ensure the $R_{max}$ values remain bounded at high intensity (large $\Delta p$). Coefficients were fitted using non-linear least squares regression. Figure 11 presents modelled and observed values of $R_{max}$ versus $\Delta p$, where the modelled values

are derived using randomly selected values of $\Delta p$ and $\lambda$. The model slightly over predicts $R_{max}$ at low intensity ($20 < \Delta p < 40$ hPa), but otherwise provides an excellent match to the observations.

## 4.4 Pressure of outermost closed isobar

The central pressure deficit $\Delta p$ used to quantify the intensity of synthetic TCs is the difference between the central pressure and the pressure of the outermost closed isobar $p_{oci}$. We initially considered the daily long-term mean sea level pressure at the location of the TC ($p_{ltm}$) as a proxy for $p_{oci}$. However, there are substantial and systematic differences between the two (Fig. 12). Using $p_{ltm}$ will lead to synthetic TCs generating sufficient wind speeds to remain defined as TCs at higher central pressure values than observed. To define $p_{oci}$ for the synthetic TCs, we modify $p_{ltm}$ based on the central pressure, latitude and day of year, plus a random innovation:

$$\boldsymbol{p_{oci} = 2324.2 - 0.65399 p_{ltm} - 1.398 p_c + 0.000740 p_c^2 + 0.00445\lambda^2 - 1.434\,cos(2\pi d_{year}/365) + \varepsilon,} \qquad (9)$$

where $p_{ltm}$ is the daily long term mean sea level pressure at the location of the TC, $p_c$ is the central pressure, $\lambda$ the latitude, $d_{year}$ the day of year. $\varepsilon$ is a random innovation sampled from a normal distribution with $\mu = 0$ and $\sigma = 2.572$. The coefficients were determined using ordinary least squares fitting to the parameters, using observed values of $p_{oci}$ from 2002-2016 JTWC records (n=1833).

Modelled values of $p_{oci}$ qualitatively match the observed values (Fig. 13), with $l^2$ norm values all less than 0.4. Closer inspection however reveals subtle differences. When plotted against $p_{ltm}$ (Fig. 13a), the maximum density of modelled values of $p_{oci}$ is skewed to lower values (approximately 3 hPa lower). For $p_c$ versus $p_{oci}$ (Fig. 13b), the comparison is much closer, with the peaks in the PDF for both modelled and observed $p_{oci}$ coinciding near weak (high $p_c$) and $p_{oci}$ near 1006 hPa. Comparison by latitude (Fig. 13c) is very good, with the peak of the PDF of modelled values overlaying the observed peak. The PDF of modelled $p_{oci}$ against day of year (Fig. 13d) is also very close to the observed distribution.

## 4.5 Landfall

Initial testing using only the autoregressive model for intensity after landfall resulted in unrealistically long-lived tracks after landfall. Instead, the filling rate of TCs after landfall is modelled in the same manner as Vickery (2005), where the central pressure deficit $\Delta p$ decreases as an exponential function of time over land $t$, the central pressure deficit at landfall $\Delta p_0$ and the translation speed at landfall, $v_0$:

$$\Delta p(t) = \Delta p_0 \exp(-\alpha t) \qquad (10)$$

where

$$\alpha = \alpha_0 + \alpha_1 \Delta p_0 + \alpha_2 v_0 \qquad (11).$$

To determine an optimum value for the parameters $\alpha_0$, $\alpha_1$ and $\alpha_2$, the decay behaviour of 174 landfalling TCs recorded in the IBTrACS dataset was analysed (Fig. 14). $\Delta p_0$ is the last observation of central pressure deficit prior to landfall, and all observations of $\Delta p$ after landfall are normalised by this value. Differences in the decay rate of TCs can be identified between those making landfall on the northwest Australian coastline and the eastern coastline (Fig. 15). We hypothesise that this is due to the presence of the Great Dividing Range along the eastern coast of Australia, with elevations exceeding 1400 metres in places (e.g. Mt Bartle Frere and Bellenden Ker). Further, the mean central pressure of landfalling cyclones in eastern Australia is higher than those along the western Australian coastline (Fig. 16). In the interest of minimising the data demands (especially with a view to application in other basins), topography was not included in the regression such that we do not have to source suitable topographic data for all potential basins. However this is an area for future development.

An exponential decay function was fitted to the normalised pressure deficit $\Delta p / \Delta p_0$ for the 174 landfalling TCs (Fig. 17). In general, $\Delta p$ follows the expected exponential decay form with $\alpha$ defined as:

$$\alpha = 0.03515 + 0.000435\Delta p_0 + 0.002865v_0 + \varepsilon(\mu, \sigma) \tag{12}$$

where ε is a random variate sampled from a lognormal distribution with μ=0.6953 and σ=0.0471, and held fixed for each event. Coefficients were fitted using non-linear least squares optimisation. This gives a decay rate parameter that is influenced by central pressure at landfall and replicates the observed decay rates well (Fig. 18). The effect of the landfall decay model can also be seen in several of the storms in Fig. 9. Storms that move back over open water revert back to the stochastic intensity model, with some storms showing reintensification. For example, track 0 makes landfall after about 220 hours, weakens, but reverts back to the stochastic intensity model near 235 hours, before a second landfall at 242 hours.

## 4.6 Lysis

Lysis of a synthetic TC occurs when $\Delta p$ falls below an arbitrary threshold, set to be 5 hPa, either due to the decline in intensity following landfall, or through the autoregressive process described above. TCs are also terminated on exiting the track domain.

## 5 Tropical cyclone wind field model

Parametric wind fields are calculated for each event in the synthetic catalogue to enable a high spatial resolution understanding of the ARI wind speeds. The additional benefit of this calculation is that users can select individual synthetic events from the catalogue and obtain a wind field for use in scenario simulations.

## 5.1 Radial wind profile

The wind field around each TC is calculated at high spatial resolution (up to 0.01°) to ensure the peak wind speeds near the eye are accurately captured. TCRM first uses a radial profile to estimate the gradient level wind associated with the vortex. To allow users to explore the range of variability in ARI wind speeds associated with different radial profiles, we have implemented a number of profiles in TCRM. These include the Holland (1980), Schloemer (1954), Willoughby and Rahn (2004), Powell et al. (2005), Jelesnianski (1966), the McConochie et al. (2004) double exponential profile and a Rankine vortex profile. The Willoughby, Schloemer and Powell et al. profiles are all variants of the Holland profile – the difference being the definition of the peakedness or β parameter. While more complex radial profiles are available in the literature, we have chosen to implement simpler models that rely only on readily available best-track parameters (e.g. central pressure, latitude). For this verification study, the Powell et al. (2005) profile was used, with β defined as:

$$\beta = 1.881093 - 0.010917|\lambda| - 0.005561R_{max} + \varepsilon \tag{13}$$

where $\lambda$ is the latitude of the TC centre and $\varepsilon$ is a random variate sampled from a normal distribution with zero mean and standard deviation 0.286. The random innovation term is held fixed for each storm event.

## 5.2 Boundary layer model

In addition to the range of radial profiles, users can also select one of three boundary layer models. These boundary layer models relate the winds at the gradient level to those near the surface, taking into account the asymmetry induced by the forward motion of the TC and surface friction effects. In parametric TC models, this is often achieved by vector addition of the forward motion and the gradient winds together with a surface wind reduction factor. Examples of this type include McConochie et al.'s (2004) model, which varies the inflow angle as a function of radial distance, or the Hubbert model

(Hubbert et al., 1991). Alternatively, a linear analytic model (Kepert, 2001) of the boundary layer flow can be applied with minimal computational cost.

In this study, the linear boundary layer model of Kepert (2001) was applied to relate gradient level winds to surface winds. This model utilises a bulk formulation for the boundary layer with the drag coefficient set to a constant value of 0.002 and the turbulent diffusivity for momentum set to 50 m$^2$ s$^{-1}$, as recommended by Kepert (2001). The model assumes $V_{tangential} \gg V_{translation}$, which may be violated for low intensity storms (e.g. incipient TCs). The boundary layer model is modified to linearly reduce the effects of translation speed when $V_{translation} > 0.2 \, V_{tangential}$. The effects are also reduced to zero at distances greater than 2 $R_{max}$, using an inverse square decay function.

The linear analytic model generates a surface wind speed corresponding to a 1-minute mean wind speed (Khare et al., 2009). This is converted to a 0.2-second gust wind speed using a wind speed conversion factor determined using the approach outlined in Harper et al. (2010). The resulting wind fields represent a 10-metre above-ground, 0.2 second gust wind speed over flat terrain with an aerodynamic roughness length of 0.02 metres. This is carried across the entire simulation domain, including over-water areas. This choice is made to enable direct comparison to other measures of regional-scale wind hazard such as weather station observations. For more localised wind speeds that can be used for detailed wind impact calculations (e.g. Krause and Arthur, 2018), local site conditions can be incorporated via an offline calculation that can incorporate local accelerations over topography and varying surface roughness conditions (Yang et al., 2014).

Throughout the simulations, it is assumed the gradient-level wind is axisymmetric. However, the simulated tracks can extend to mid-latitudes, where TCs undergo transition to extra-tropical cyclones and the gradient level wind becomes asymmetric (Foley and Hanstrum, 1994; Jones et al., 2003; Loridan et al., 2013). Further, the assumption in the linear boundary layer model that $V_{tangential} \gg V_{translation}$ does not hold for transitioning storms, where $V_{translation}$ can exceed 70 km/h (Foley and Hanstrum, 1994). This means the simulated hazard values in the mid-latitudes (approximately poleward of 30° in the south-eastern Indian Ocean) are likely not indicative of the true wind hazard associated with (transitioning) TCs. It is also likely in these regions that other phenomena (e.g. thunderstorms) are the predominant source of extreme wind gusts. There are promising developments in the area of extra-tropical transition (Loridan et al., 2015; Bieli et al., 2019), which have direct application in probabilistic modelling frameworks and may be integrated into TCRM in future releases.

## 6. Extreme value distribution fitting

Once wind swaths for the simulated TCs have been generated using the wind field module, the maximum wind speed from all simulated events, irrespective of direction, for each grid point is stored. Because of the large number of events simulated, it is possible to estimate average recurrence intervals (ARIs) for the wind speeds at each grid point. The simplest approach is to use an empirical approach based on Eq. (14):

$$ARI = \frac{n_{obs}}{1 - \frac{r_i}{n_{events}+1}} \tag{14}$$

where $r_i$ is the wind-speed rank of the $i$th event, $n_{obs}= 365.25$ is the number of values per simulated year and $n_{events}=10000 \times n_{obs}$ is the total number of simulated 'daily' observations for the 10,000 year simulation. For each point across the simuation domain, we treat each simulated event as an individual 'daily' observation and rank the simulated wind speeds from all events. This usually produces around 10$^5$ records (depending on the frequency of TCs at that location). The remaining 'daily' records are zero filled.

A more sophisticated approach is also implemented, where the simulated maximum wind speed values are fitted to a Generalized Pareto Distribution (GPD) using a peaks-over-threshold approach. ARI wind speeds are estimated from the GPD parameters using Eq. (15):

$$w(t; \mu, \sigma, \xi) = \mu + \frac{\sigma}{\xi} \left[ (n_{obs} \rho t)^\xi - 1 \right] \tag{15}$$

where $w$ is the wind speed with an ARI of $t$ years. $\mu$, and are the location, scale and shape parameters of the fitted GPD distribution respectively and is the rate of exceedances above the threshold u. The threshold is set to the 99.5th percentile of the simulated wind speed values. As wind speeds are considered a bounded phenomenon (Lechner et al., 1992), the fitted shape parameter can be constrained to be positive to ensure the resulting distribution is bounded at long return periods (Holmes and Moriarty, 1999). Again, this parameter fitting is performed at each point across the region of interest, leading to a spatial representation of ARI wind speeds.

Confidence intervals are estimated from the covariance matrix of the parameter fit. This method is useful for estimating winds speeds at very long ARIs, where the frequency of events is very low. ARI wind speeds estimated using this method tend to be underestimated at lower ARIs compared to the empirical approach, largely because the threshold selection excludes lower, more frequent wind speeds.

# 7 Results

## 7.1 Track model verification

To evaluate the performance of the model, we run a series of comparisons between the observed tracks and a large number of simulated track sets. 1000 synthetic event sets were generated, each representing 35 years of TC activity, mimicking the length of the input historical record. For each metric, historical values are compared to the mean value for the collection of synthetic event sets, with 90th percentile confidence intervals calculated using bootstrapping methods.

The distribution of observed longitude crossing rates are well modelled for both eastward- and westward-moving storms (Fig. 19). The values represent the probability density of events crossing each longitude in 2 degree latitudinal segments. In general, the model simulates the longitude crossing rates well. Near 150°E, 15°S the model does not capture the rate of TCs crossing Cape York Peninsula, which is related to the termination of TCs due to low intensity. Similar overall results are obtained for longitude crossing rates when the model is tested in the western North Pacific and Atlantic basins (not shown), including in mid- to high-latitudes, capturing the paths of recurving TCs.

TCRM simulations of minimum central pressure perform well, especially for the more intense (lower minimum central pressure) TC events (Fig. 20). The lower tail of the simulated distribution closely follows the observed tail. For weak TC events (> 980 hPa), the observed distributions lie outside the 90th percentile of the simulations. This has little impact on the derived extreme wind speeds, which are generated by the most intense TCs.

The spatial distribution of minimum central pressure is presented in Fig. 21. Values represent the lowest minimum central pressure value observed in each 1° by 1° grid cell in the historical record. We apply the same process to each of 1000 synthetic event set, and the mean of those is presented. There is general agreement between the synthetic and observed (historic) distributions, though individual events in the historic record do result in greater variability. The spatial distribution of the mean central pressure (Fig. 22), calculated in a similar manner to the minimum central pressure, again shows good agreement between the synthetic and observed event sets, without the large variability seen in the historic minima.

Lifetime maximum intensity (LMI), defined as the maximum wind speed for the life of the TC, can be used to evaluate how well the model simulates the evolution of intensity of TCs. Figure 23 shows the mean location of LMI in the observed (top) and simulated (bottom) records. There is little discernible pattern in the location of LMI for observed TCs. This may in part be due to the small numbers of events available for the analysis (n=377), as only those events with a maximum wind speed recorded were used. The mean LMI is generally evenly distributed throughout the domain, though lower LMI values can be seen over Cape York (145°E) and south of Indonesia at low latitudes. For the simulated events, there is a clear trend towards higher LMI at higher latitudes in the Indian Ocean, with highest LMI simulated at 20-25°S. There is also indications that LMI increases towards the south in the Coral Sea to the east of Australia.

The average time taken to achieve LMI for observed TCs (Fig. 24, top) shows little clear pattern, though the areas off the western coast of Australia do tend to be slightly higher. The simulated tracks (bottom) display a strong tendency to achieve LMI at higher latitudes (> 17.5°S), and take longer to achieve LMI in these areas. This may lead to higher ARI wind speeds at these higher latitudes. Comparing geographical areas, observed time to LMI off the northwest coastline is around 96-144 hours, while the mean time to LMI is only around 48-72 hours for simulated events. On the east coast, there appears less difference between observed and simulated time to LMI, but there is a greater range of values, ranging between 48 and 120 hours.

Figure 25 presents the landfall probabilities around the Australian coastline. Each gate is 200 km wide and located approximately 50 km off the coast (Fig. 1). TCRM replicates the observed probability of landfall well, with the mean of the synthetic event sets generally close to the observed probability. The relatively low occurrence of landfall around the Australian coastline (on average only 4 TCs cross the coast each year) means there is large variability in the landfall count for any given synthetic event set. Again, qualitatively similar results are obtained for simulations in the western North Pacific and Atlantic basins (not shown).

At all times the 80th percentile range captures the observed landfall probability, as expected. However, the mean landfall probability in the simulations for the region between Coral Bay and Port Hedland is substantially lower than the observed landfall probability. This is possibly linked to lower genesis probabilities directly to the north of this area. Historically, there is a local maximum in genesis probability between 120°E and 130°E, extending westward into the Indian Ocean near 10°S (Fig. 5). The mean genesis density for the simulations does not show the same local maximum, or the westward extension. This lower genesis density is likely translating into lower track densities in the region, and therefore lowers landfall rates along the northwest Australian coast. This in turn acts to reduce ARI wind speeds along this part of the coastline compared to observed ARI wind speeds.

When examining the distribution of intensity at landfall (Fig. 26), the proportion of category 5 landfalls along the lower west coast is high, representing around 25% of landfalls through gates between Cape Leeuwin and Coral Bay. This is hypothesised to be linked to the poor representation of transitioning TCs in this region. The region from Coral Bay to Port Hedland (noted previously for a low landfall probability) displays a similar landfall intensity distribution to observations, where the majority of landfalling TCs are severe (category 3-5). Along the east coast, between Mackay and Coffs Harbour, the proportion of category 5 landfalls is also high (around 20%). Additionally, the probability of any landfall in this region is significantly higher than the observations. These two factors contribute to an overestimate of the ARI wind speeds in southern Queensland and northern NSW.

## 7.2 Wind field model verification

To demonstrate the reliability of the wind field model, we modelled a number of historical TC events using the parametric wind field and compared to observed wind speeds recorded at Bureau of Meteorology weather stations near to the track

of the TC. We select stations that are within 100 km of the track, in an effort to verify against the strongest winds of the TC and provide a meaningful result. In total, 29 stations and 14 TCs are examined, providing a cursory analysis of the wind field model performance.

The configuration is kept consistent with that used to derive the ARI wind speeds, so no calibration of the parameters (e.g. peakedness parameter) is performed for individual TCs used in the verification (c.f. McConochie et al., 2004). We chose not to calibrate for each individual event so as to quantify the capability of the wind field model to reproduce gross features of observed TCs. In a stochastic model such as TCRM, it is important that the wind field model display no significant bias in wind speed, and errors in the mean wind field are minimised. By using a consistent configuration for verifying the wind field model, it is possible to quantify the bias in the wind field that might arise in applying that configuration to a set of synthetically generated TCs. This will generally result in a poorer simulation of each individual TC when assessed using metrics such as root mean square error, since the parameters have not been optimised for the evolution of each individual event. The reader is directed to the references for the profiles and boundary layer model for more thorough validation of those models.

Simulated wind speeds are matched to corresponding observations from weather stations based on the time of observation. The wind field model provides data at five minute intervals, so it is possible that absolute peak observed values may not match the time interval. Observed wind speeds are corrected for gust averaging time periods (Harper et al., 2010), as the default configuration for TCRM is to produce wind speeds representing a 0.2 second gust wind speed. No corrections for site exposure (e.g. topographic enhancement, surface roughness changes) are made.

### 7.2.1 Weather station wind histories

For each simulated event, time histories of wind speed and direction at all weather stations within the modelled domain are recorded. Figures 27 to 30 present the time history for four stations (Mardie, Carnarvon, Lucinda Point and Flinders Reef) during the passage of four separate TCs (*Glenda*, *Olwyn*, *Yasi* and *Larry* respectively). For each of these events, the TCRM wind field simulation captures the increase in wind speed as the TC approaches the station, with the time of peak wind speeds accurately modelled.

Changes in wind direction follow the observations closely, except for Lucinda Point (Fig. 29) where the observed winds quickly returned to a southerly direction after the passage of TC *Yasi*. The simulation shows winds turning anti-clockwise as the TC passes at around 00Z on February 3 2011, consistent with a cyclonic vortex passing north of the observation site. It appears a similar shift occurs in the observations several hours earlier, but winds turn back to the south rapidly at around 18Z on February 2. This difference is likely due to the model not containing any other sources of pressure gradient winds from synoptic-scale weather patterns such as high pressure ridging into the Coral Sea following the passage of the TC (for example, see the discussion in McConochie et al., 1999).

Peak wind speeds are closely simulated, except for Lucina Point in TC *Yasi*, where the wind field model underestimates the peak wind speed by nearly 20 m s$^{-1}$. Among other events (Fig. 29, lower right panel), the tendency is for the TCRM wind field simulation to overestimate peak wind speeds. There are likely several factors (for example, instrument failure or site exposure) leading to this overestimation, which may be drawn out in a more thorough validation of the wind field model and analysis of those event. The results here are likely due to our decision not to correct the observed wind speeds for site exposure, which would reduce the modelled wind speeds.

For the complete time histories (all corresponding time records, not just the peak), Fig. 31 presents the root mean square error, bias and mean absolute error for all 29 stations and 14 events modelled. The average RMSE is 9.2 m s$^{-1}$ and the bias 1.2 m s$^{-1}$. In the context of a stochastic model where many thousands of events will be simulated, the average RMSE outcome is acceptable, but the tendency for peak wind speeds to be overestimated requires further investigation.

## 7.3 ARI wind speed verification

ARI wind speeds for the Australian region are calculated from a simulation of 10,000 simulated TC seasons, or over 160,000 individual simulated TC events. Two ARI wind speeds are examined – the 50-year ARI wind speeds are compared to observed TC-related wind speeds, while the 500-year ARI wind speed is compared to the regional design wind speeds detailed in AS/NZS 1170.2 (2011). We use the 500-year ARI wind speed for comparison, as it represents the regional design wind speed for residential housing in AS/NZS 1170.2 (2011).

Observed ARI wind speeds were estimated from daily maximum gust wind speed observations that may be attributable to the passage of a TC, recorded at Bureau of Meteorology weather stations (Fig. 1). All TCs passing within 200 km of the station, when the station was open, were recorded. Daily maximum wind gusts corresponding to the closest passage of each TC were then extracted and empirical ARI values determined based on Eq. (14). Corrections are made for gust wind speed averaging times where instrumentation is known (Harper et al., 2010).

The 500-year ARI wind speed map (Fig. 32) displays qualitative similarities to existing design wind loading standards (see Fig. 3.1A in AS/NZS 1170.2), at least over continental Australia (the design standard does not define wind speeds over the ocean). Highest wind speeds are estimated along the northwest coast of Australia, with a peak value near 70 m s$^{-1}$ near Port Hedland (near 120°E) for the 500 year ARI. The remainder of the northern and much of the eastern coastline indicates lower wind speeds, generally between 50 and 60 m s$^{-1}$. The wind speeds drop below 45 m s$^{-1}$ across Cape York (142°E), where there is also a marked decrease in TC frequency. The highest values of 500-year ARI wind speed along the east coast occur around the Rockhampton region, reaching 60 m s$^{-1}$. ARI wind speeds steadily decline further south, reaching around 50 m s$^{-1}$ in northern New South Wales. The comparatively high values at higher latitudes (c.f. AS/NSZ 1170.2) are likely indicative of the model failing to correctly simulate the extratropical transition process noted in section 5.2.

ARI curves for selected locations are presented in Fig. 33, along with estimated ARI wind speeds for observed TC-related wind speeds (see below) at those locations. The solid line is a GPD fitted to the simulated wind speeds using peaks-over-threshold with a 99.5$^{th}$ percentile threshold, but here we have relaxed the constraint of $\xi > 0$. 90$^{th}$ percentile confidence intervals are determined from the covariance matrix of the fitting routine.

For locations along the west coast (Carnarvon, Port Hedland), the model underestimates the hazard profile compared to the observed hazard profile. For Port Hedland (64 years of records), the simulated 50-year ARI wind speed is just under 50 m s$^{-1}$, while the observed 50-year ARI is approximately 57 m s$^{-1}$ (Table 1). Similarly at Carnarvon (65 years of records), the simulated 50-year ARI is 35 m s$^{-1}$ and the observed 50-year ARI is close to 45 m s$^{-1}$. This is a significant discrepancy between the simulation and observations. In part, this is attributed to the comparatively low landfall rates along this section of the coastline (Fig. 25, 26), and so points to additional work on improving the simulation of intense TCs along this section of the Australian coastline, in terms of both event rates and intensity.

In other parts of the country, model performance is much better. Simulated ARI wind speeds at Darwin closely match the observed ARI wind speeds, but the outlying observation of TC *Tracy* (1974 – 67 m s$^{-1}$) is cause for further investigation. This analysis places the observed wind speed of *Tracy* at around a 5000-year ARI, but there is significant uncertainty on this estimate, and conjecture on the most appropriate way to estimate this from observations (Harper et al., 2012). For east coast locations, the simulated ARI wind speeds for Cairns, Townsville and Rockhampton are all close to the observed ARI wind speeds, varying at the 50-year ARI by at most a few percent. The observed ARI wind speeds generally fall within the 95$^{th}$ percentile confidence interval of a GPD fitted to the simulated values (not shown). It is also notable that observed and simulated ARI wind speeds lie below the corresponding regional wind loading design levels specified in AS/NZS 1170.2 (2011) for all locations, except for Port Hedland where observed ARI wind speeds are close to the regional design level.

An important component of stochastic models is to check for convergence in solutions (Shome et al., 2018). For TCRM, this can be checked by splitting the synthetic catalogue into two subsets, calculating ARI values from each and examining the range of values. A large difference in ARI wind speeds indicate the model has not converged. It is expected that at large ARIs the model would not converge, as variability in the tail of the distribution is to be expected when modelling rare events. Significant difference in the ARI values for the two subsamples indicates the variability in the distribution is large. Figure 34 shows the convergence checks for the locations mentioned above. Carnarvon, Port Hedland and Willis Island show little difference in the subsets below the 1000-year ARI level – generally differing by less than 2%. Darwin, Townsville and Broome show divergence in the subsets at around the 500-year ARI level. Cairns and Rockhampton both display weak convergence beyond the 100-year ARI level, but converge again above 1000 years. These results suggest that it may be required to run larger catalogues to achieve robust convergence of the ARI values, in line with other hurricane catastrophe models (Shome et al., 2018).

## 8 Conclusion

The Tropical Cyclone Risk Model, developed at Geoscience Australia, is a new statistical-parametric model of TC behaviour that is capable of delivering a high-resolution (approximately 2 km) spatial understanding of ARI gust wind speeds due to TCs, at continental scales. It is a free and open-source software model, with the goal of delivering TC wind hazard information to the hazard and risk modelling community in a free and transparent manner. Potential applications include evaluation of risk, scenario modelling for emergency management planning, informing wind loading requirements for building standards and projections of future climate hazard and risk. The model provides information that can readily be used to guide other hazard assessments, such as wave climate modelling and coastal storm surge, and there is potential to include other perils such as rainfall through appropriate parametric models (Lonfat et al., 2007; Mudd et al., 2015).

Initial evaluation of the model was performed using historical best-track data for the Australian region to generate a catalogue of 10,000 years of events. The statistical track model performs well in areas with a high density of TC events, but confidence is reduced at higher latitudes and near the equator due to the lower number of historical events. Simulated ARI wind speeds are generally consistent with observations of TC-generated winds around Australia, except in the northwest of the country where the simulated ARI wind speeds are significantly lower than observed. This deficit is linked to low landfall rates in this part of the country and will be investigated into the future. There are also significant opportunities to improve model performance at mid-latitudes, especially in processes such as extra-tropical transition, where other types of weather phenomena may have a substantial influence on the wind hazard climate.

## Code availability

The Tropical Cyclone Risk Model code described in this manuscript is available at http://github.com/GeoscienceAustralia/tcrm/ releases/tag/v2.1. Code used in the analysis is available on request from the author.

## Competing interests

The author declares no conflict of interest.

**Acknowledgements**

The author gives thanks to the Australian Bureau of Meteorology for making available the best track data for the southern hemisphere and daily wind observation datasets. The simulations were completed using the resources of the National Computational Infrastructure (NCI) Facility, which is supported by the Australian Government. The manuscript benefitted greatly from the contributions of Gareth Davies and Claire Krause and two external reviewers. This work has been in part supported by the former Department of Climate Change and Energy Efficiency. This paper is published with the permission of the CEO, Geoscience Australia.

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

**Table 1: Observed and simulated 50-year ARI wind speeds for selected locations. Simulated ARI wind speeds are taken to be the empirical ARIs. All values are in m s⁻¹.**

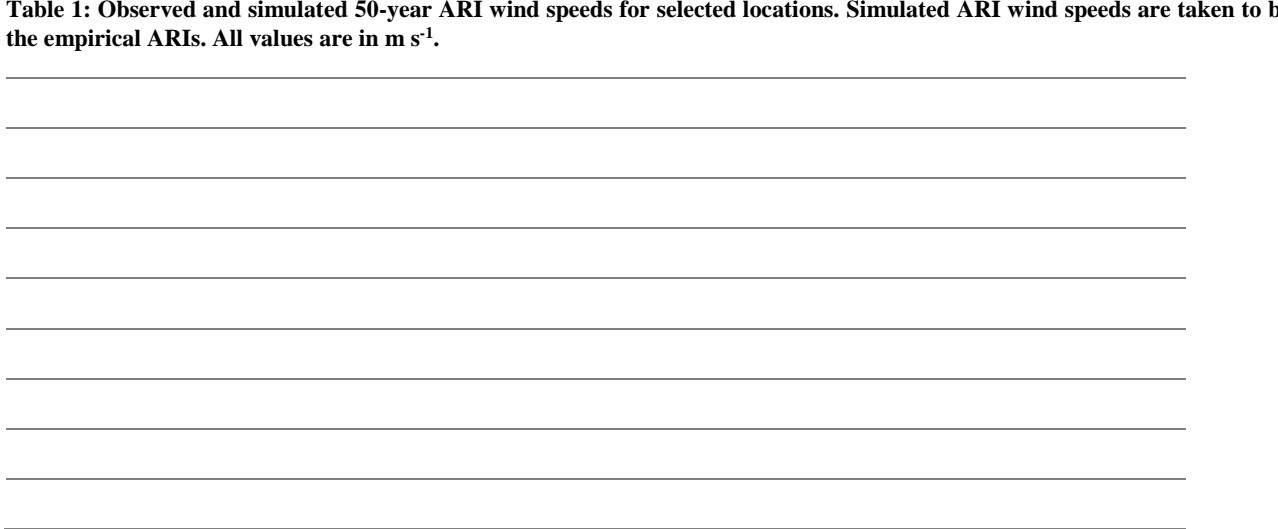

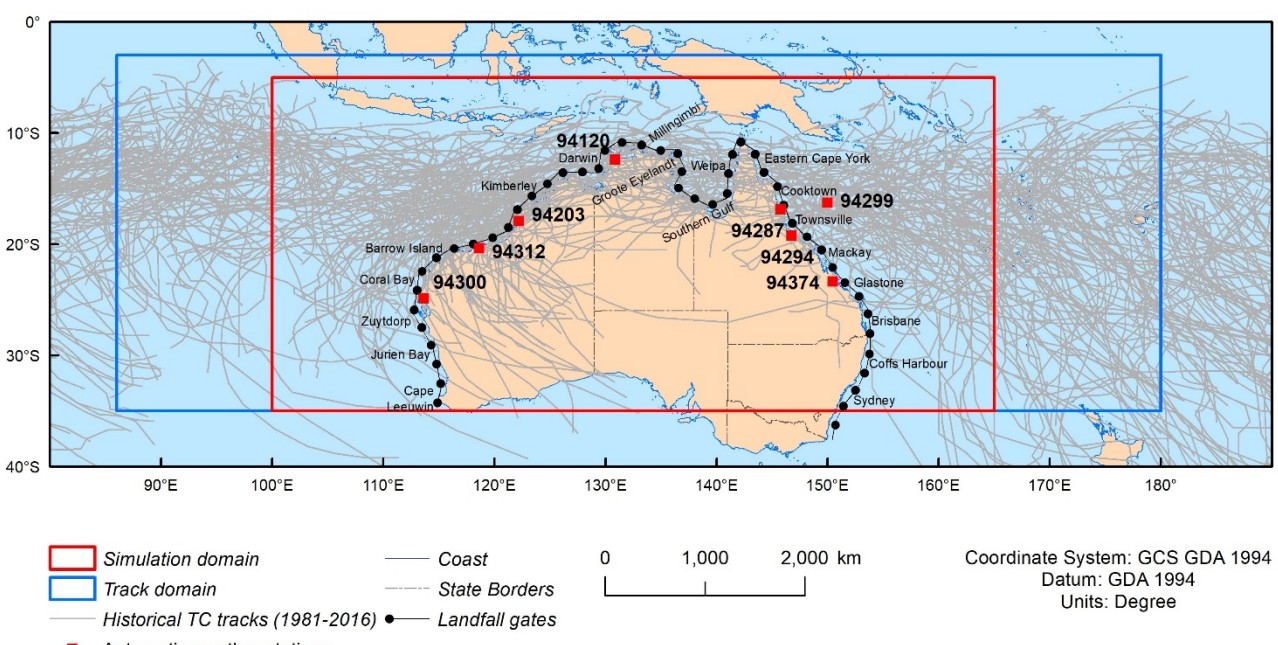

**Figure 1: Historical TC tracks (1981-2016), simulation domain, track domain, automatic weather stations (with station number) and coastal gates (50 km offshore, 200 km wide) used for landfall analysis (selected gates labelled).**

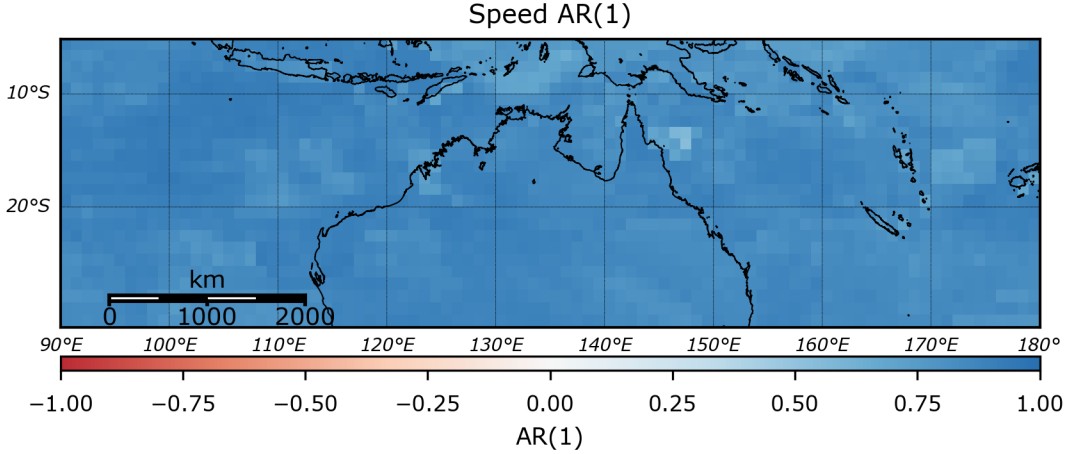

**Figure 2: Lag-1 autocorrelation of TC translation speed, based on IBTrACS v03r09 (1981-2016).Values are calculated on a 1x1 degree grid.**

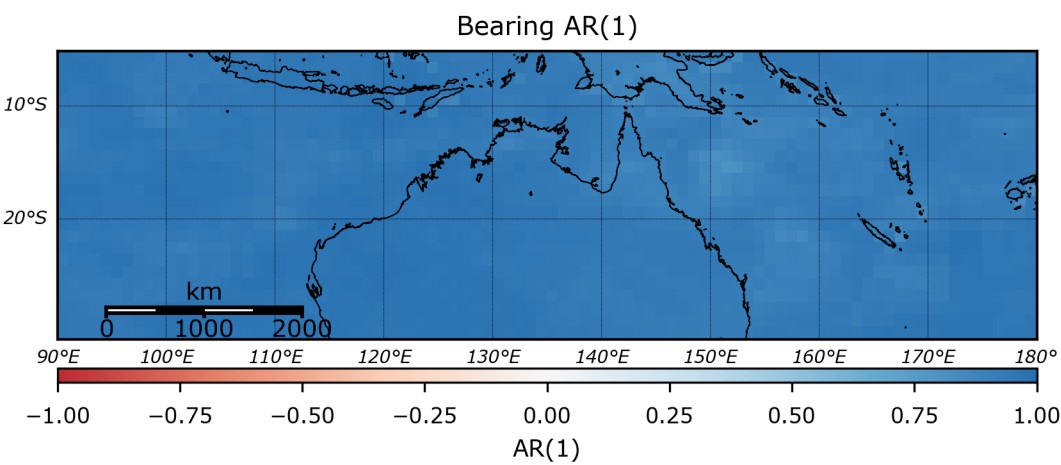

**Figure 3: Lag-1 autocorrelation of TC bearing (direction of movement), based on IBTrACS v03r09 (1981-2016).**

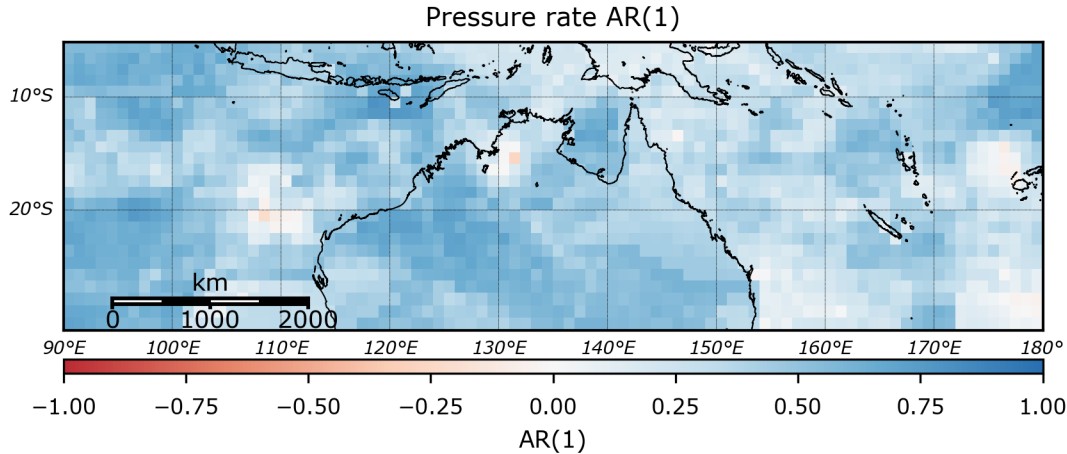

**Figure 4: Lag-1 autocorrelation of central pressure rate of change, based on IBTrACS v03r09 (1981-2016).**

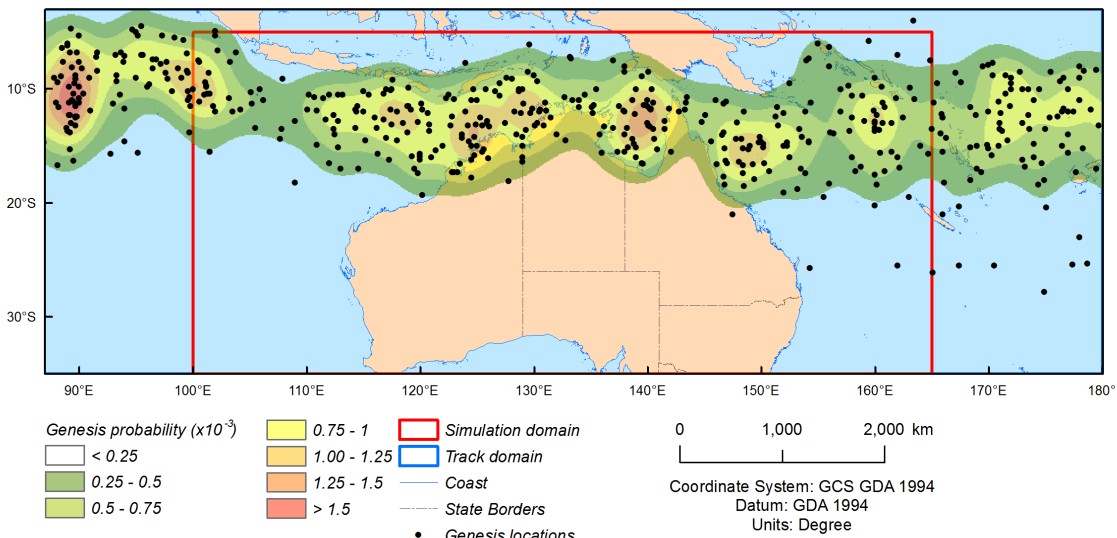

**Figure 5: TC genesis points for historical TC events (1981-2016), and the corresponding probability density function (TCs/year). Points do not represent first observation of TC intensity – rather, they represent the first point recorded in the best-track database.**

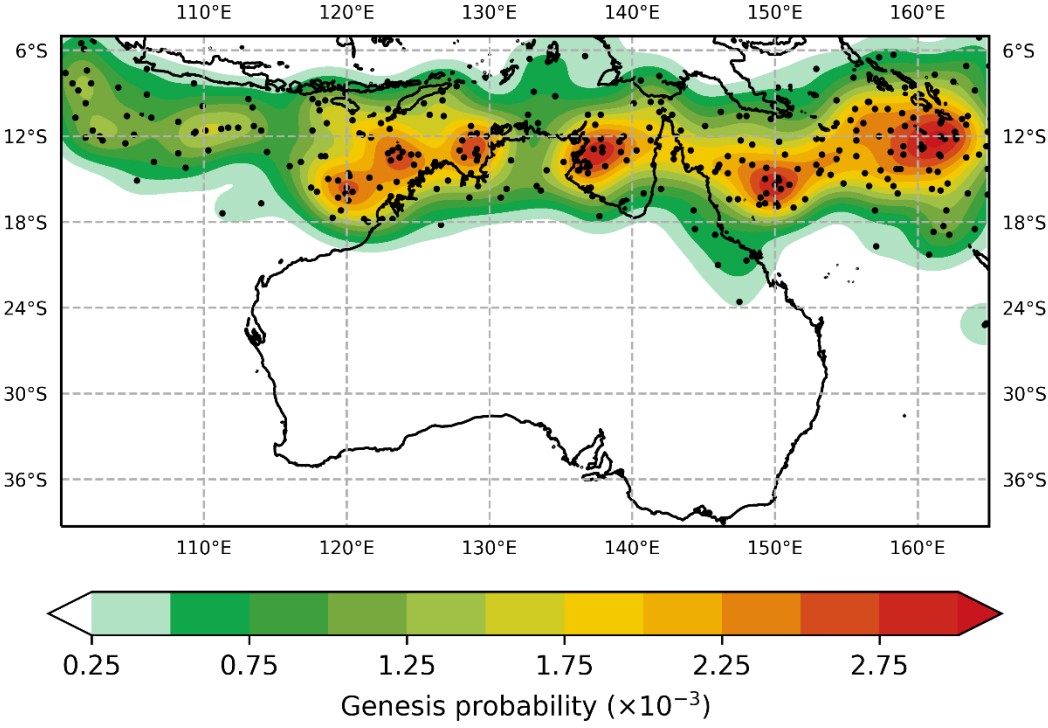

**Figure 6: TC genesis points and corresponding probability density function for a sample of 35 simulated years of TC activity. Note the different scale compared to Figure 5.**

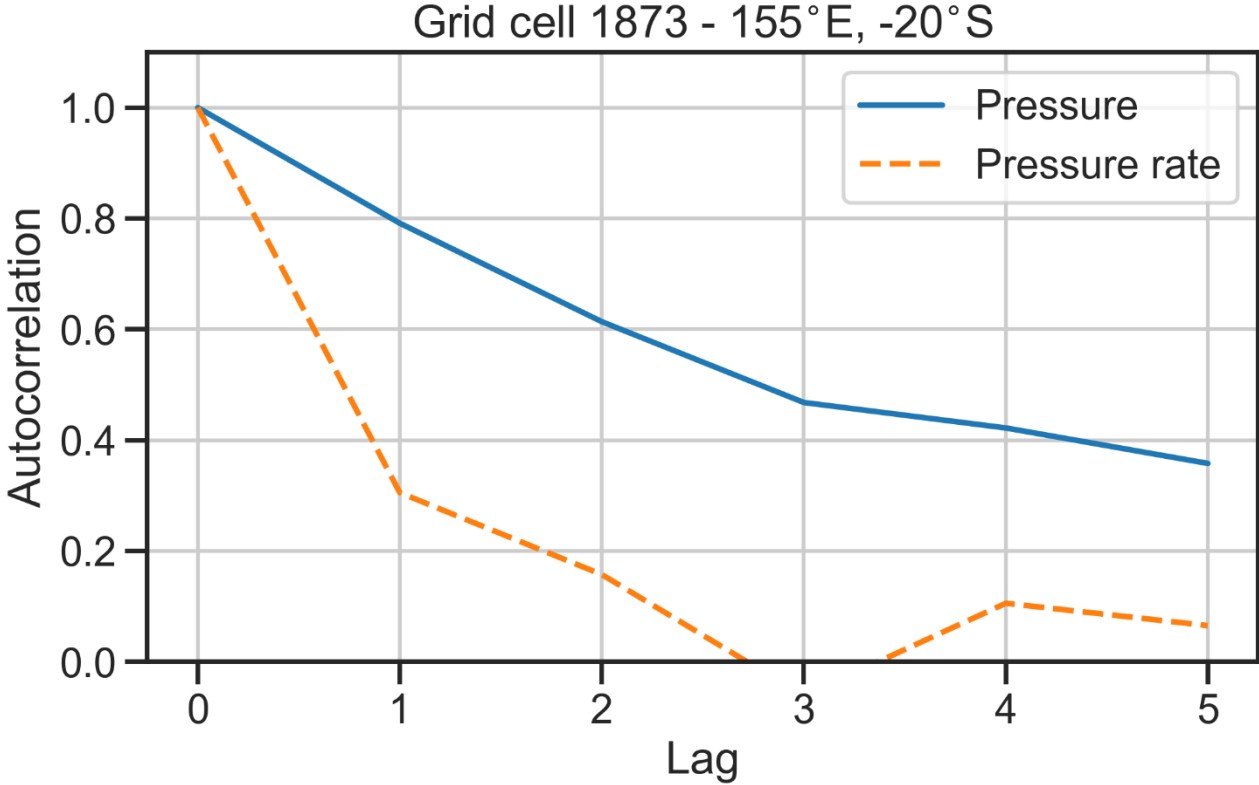

**Figure 7: Autocorrelation values for minimum central pressure and pressure rate of change, for a single grid cell in the Coral Sea.**

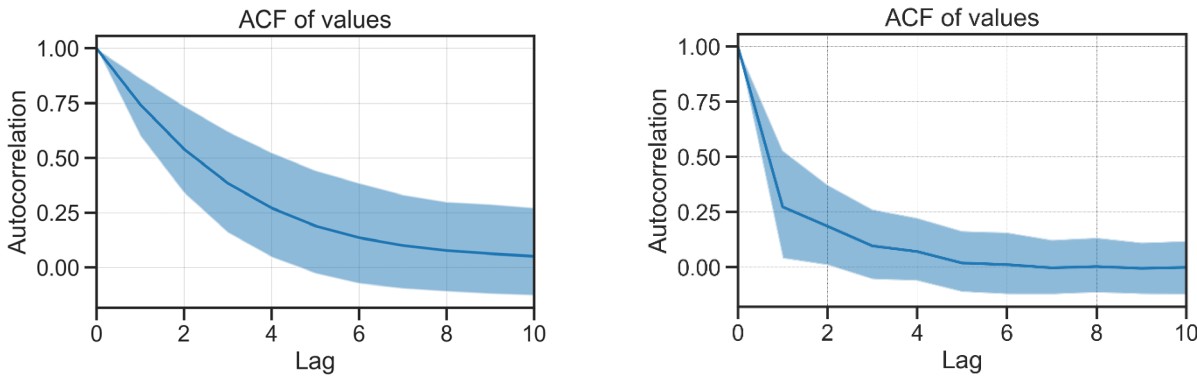

**Figure 8: Autocorrelation of minimum central pressure (left) and pressure rate of change (right) for lagged observations between 1 and 10 steps. The solid blue line is the mean for all grid cells in the track domain, and the shading is one standard deviation.**

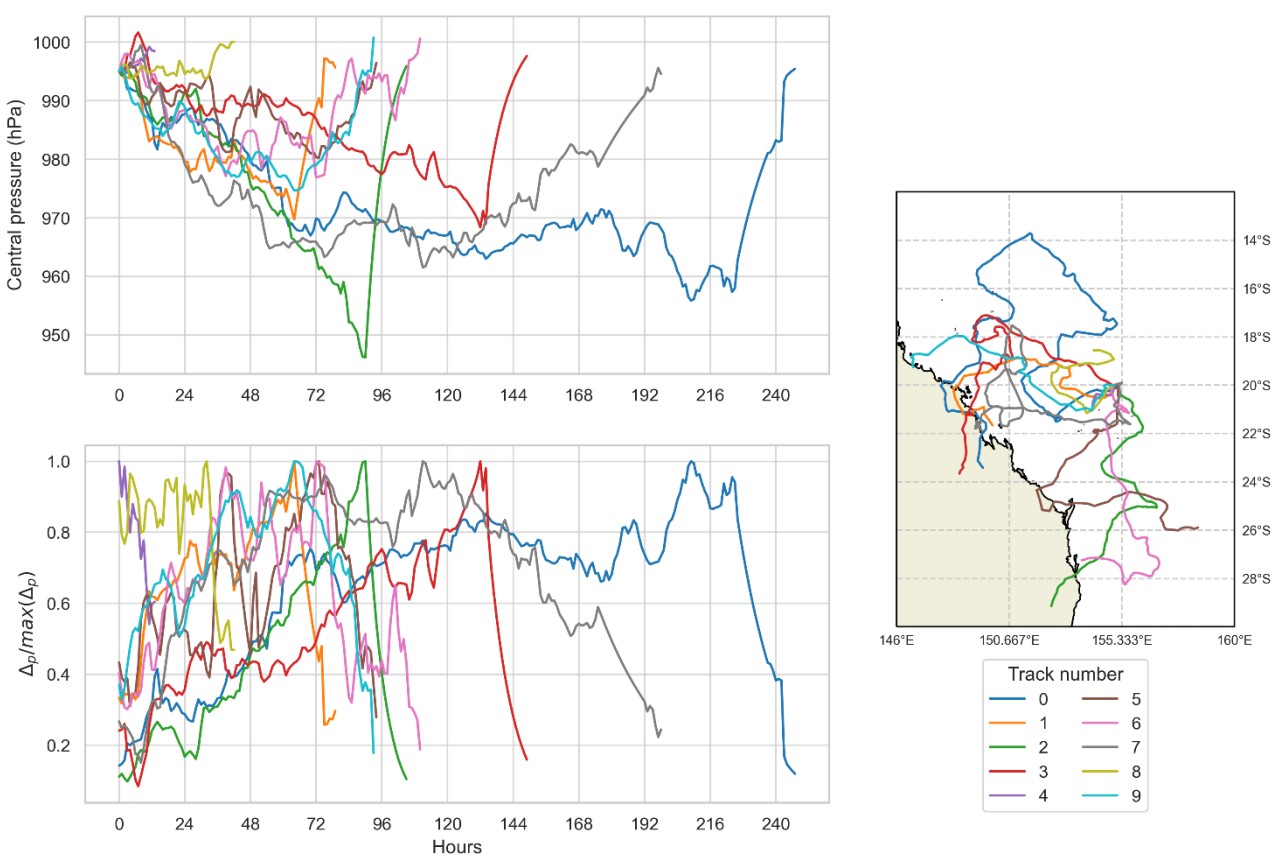

**Figure 9: Central pressure, normalised intensity and tracks of 10 events with a common genesis point (155°E, 20°S) and initial intensity (995 hPa). The lower left panel presents the normalised intensity Δp/max(Δp). Colours are for clarity only.**

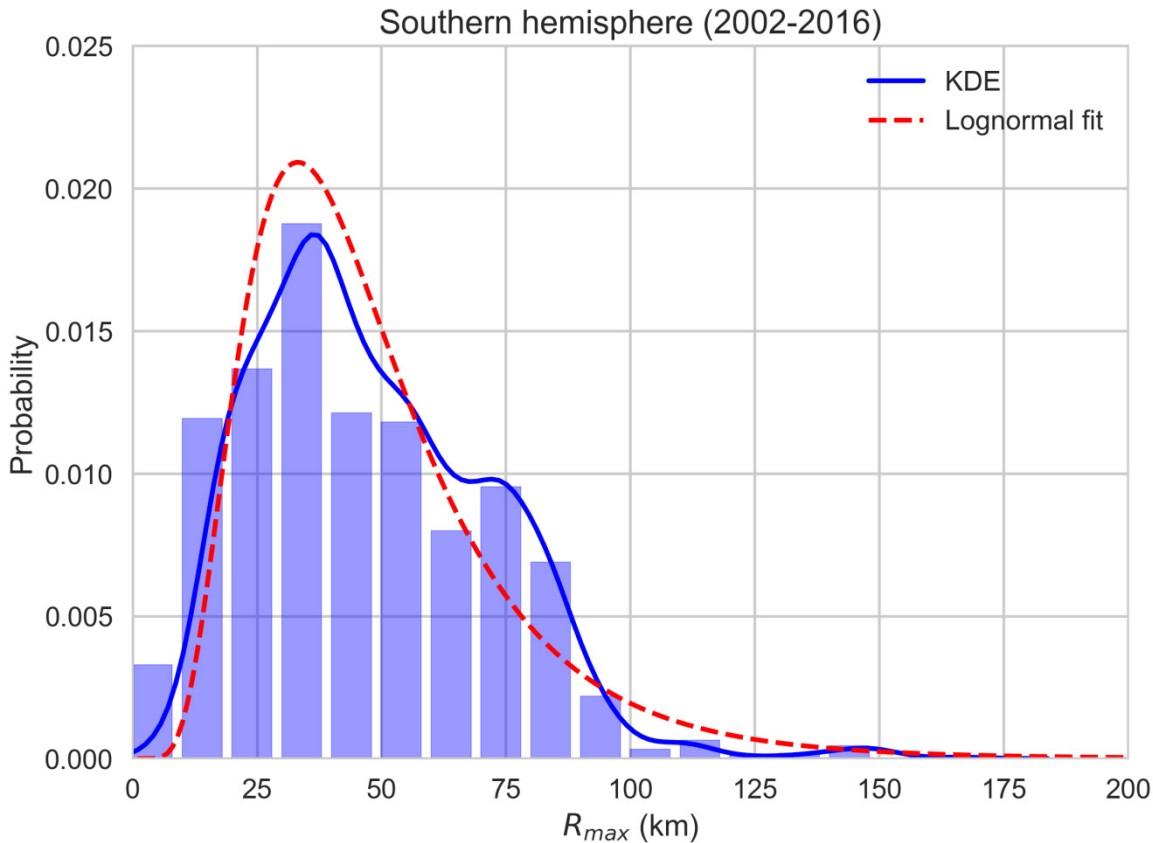

**Figure 10: Distribution of radius to maximum wind for all Southern Hemisphere TCs (2002-2016). 'KDE' is the empirical distribution determined using kernel density estimation, 'Lognormal fit' is a fitted lognormal distribution using maximum likelihood estimation. Data source: JTWC (2017).**

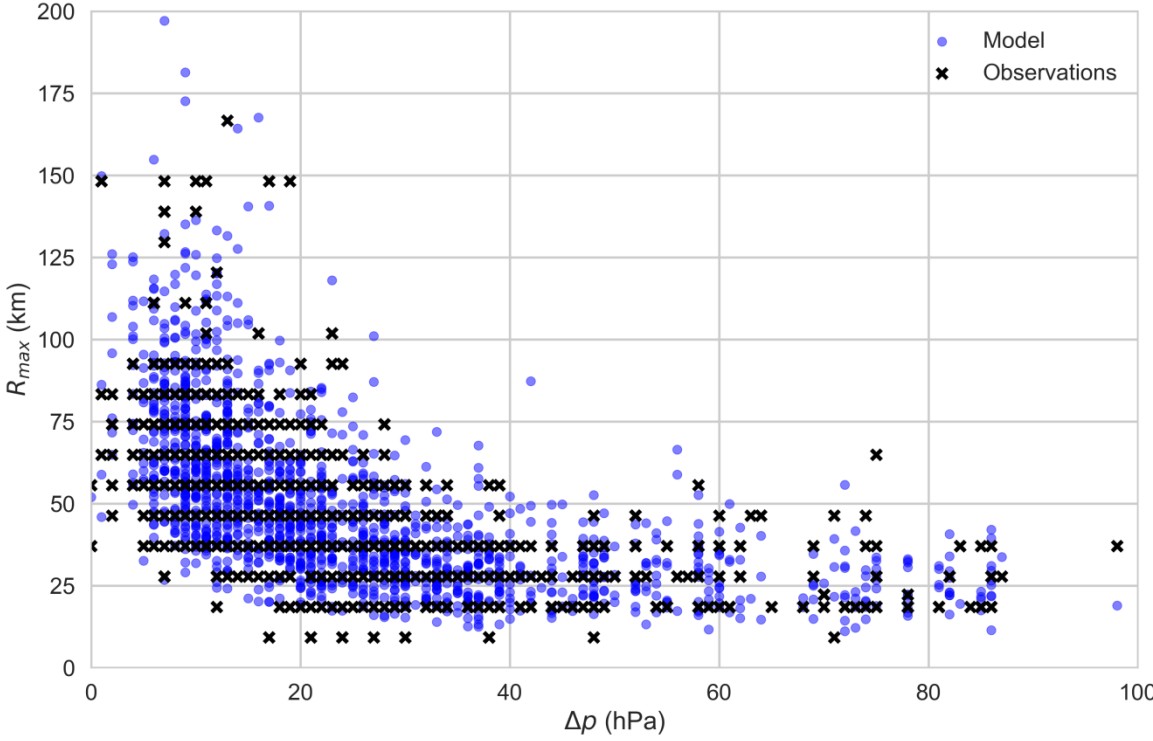

**Figure 11: Modelled (o) and observed (x) for southern hemisphere TCs (2002-2016), plotted against central pressure deficit. Modelled values are based on a random selection of observed combinations of central pressure deficit and latitude.**

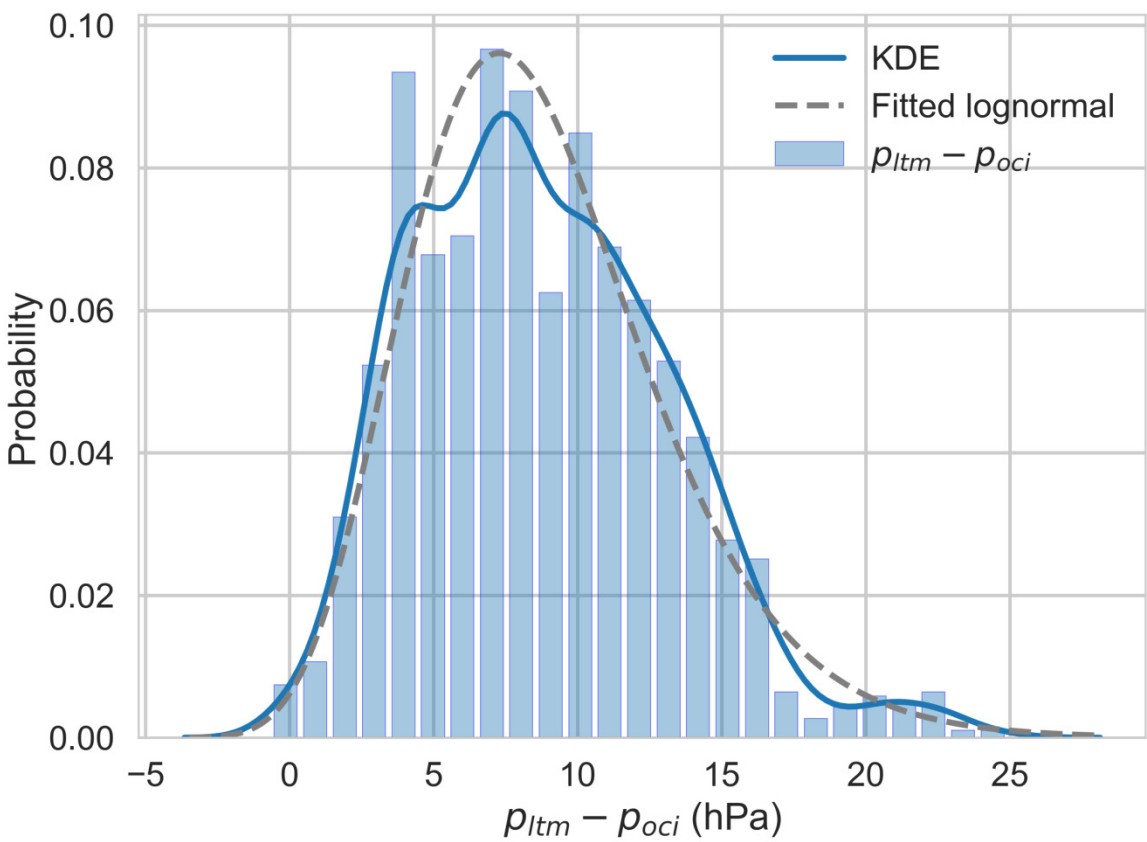

**Figure 12: Difference between      and      for southern hemisphere TCs (2002-2016). Solid line is the empirical distribution determined using kernel density estimation. Dashed line is a fitted lognormal distribution. Data source: JTWC (2016).**

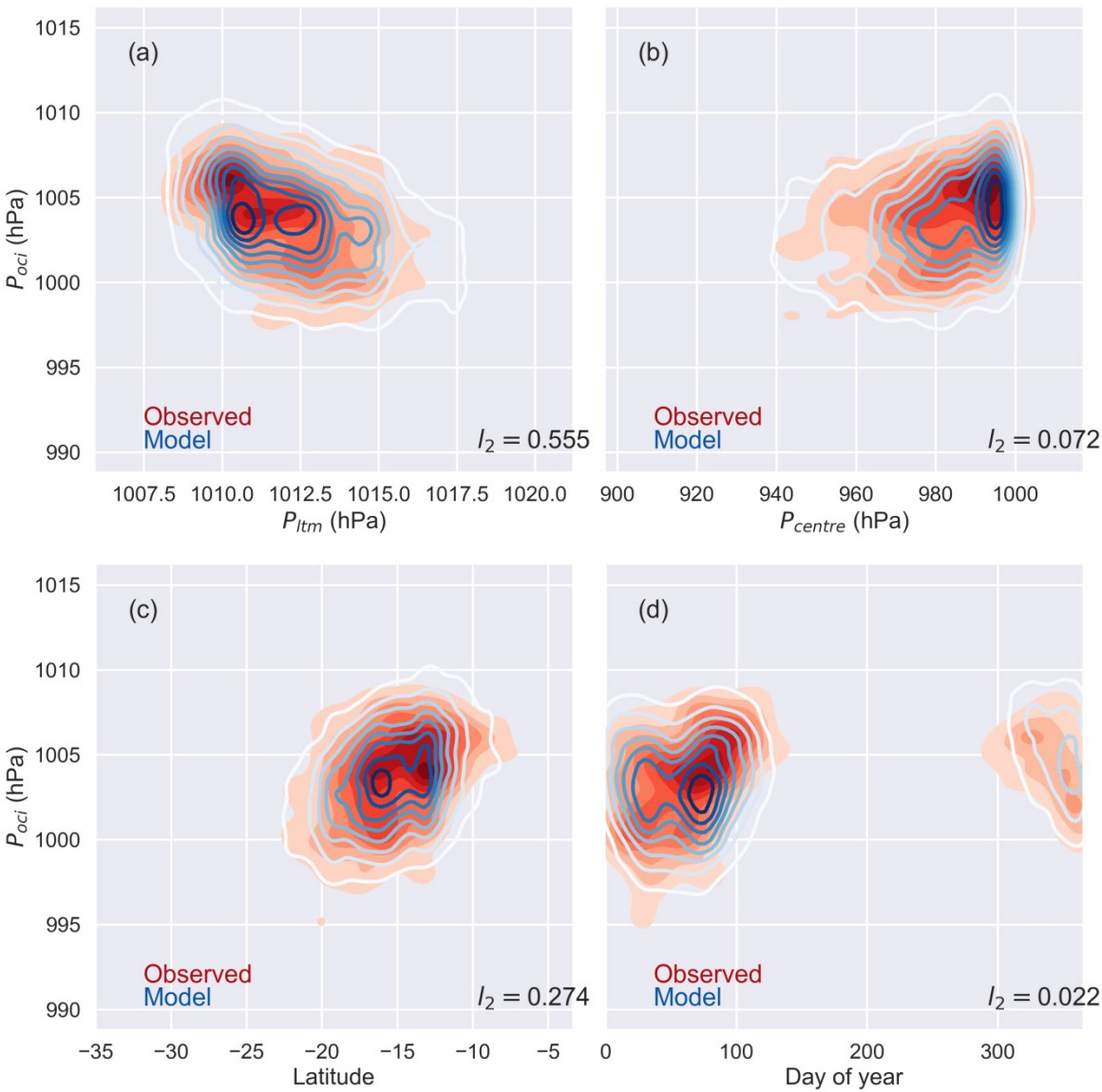

**Figure 13: Observed (shaded) and modelled (contours) distribution of       for southern hemisphere TCs, plotted against (a)      , (b)    , (c) latitude and (d) day of year. Modelled values are based on a random selection of observed combinations of    ,    , latitude and day of year. Contour interval is 0.01. Note the different horizontal scale in each panel.**

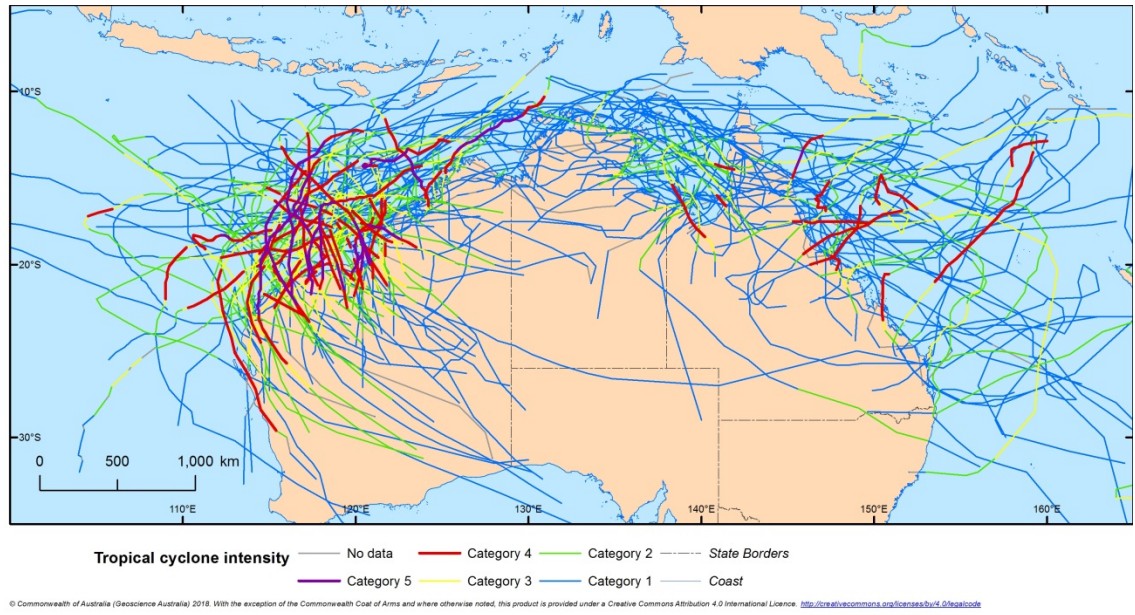

**Figure 14: Landfalling tropical cyclones in Australia, 1981-2016. Source: IBTrACS v03r09 (1981-2016).**

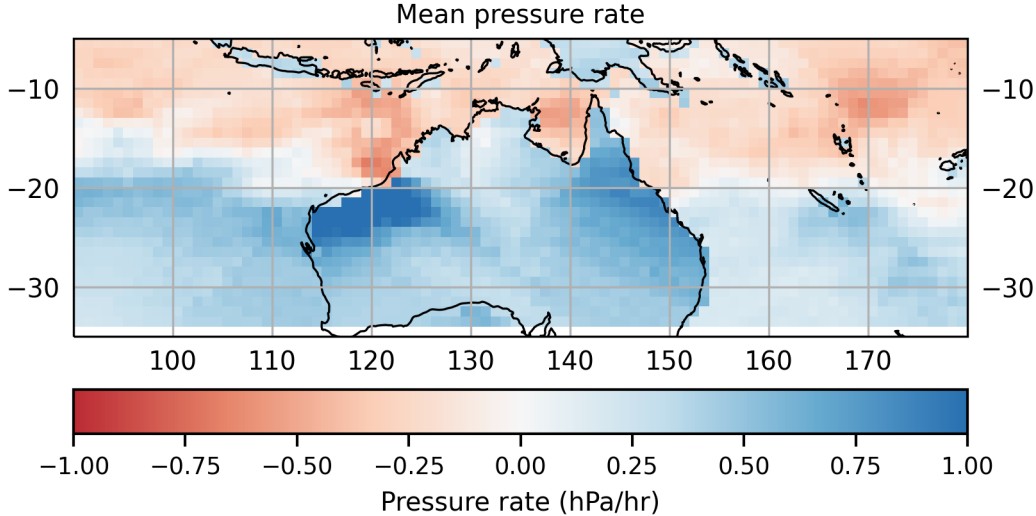

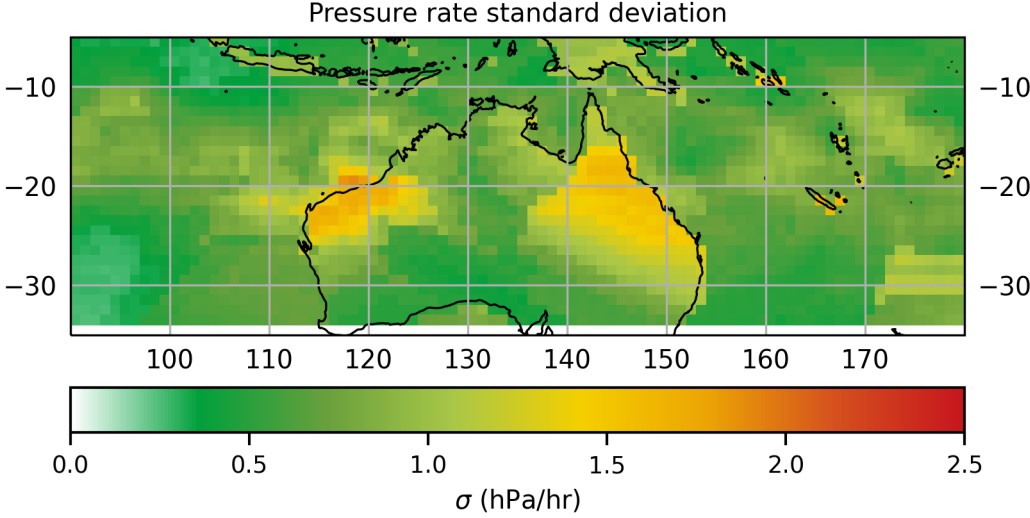

**Figure 15: Mean (top) and standard deviation (bottom) of rate of change of central pressure (hPa/hour), based on IBTrACS v03r09 (1981-2016).**

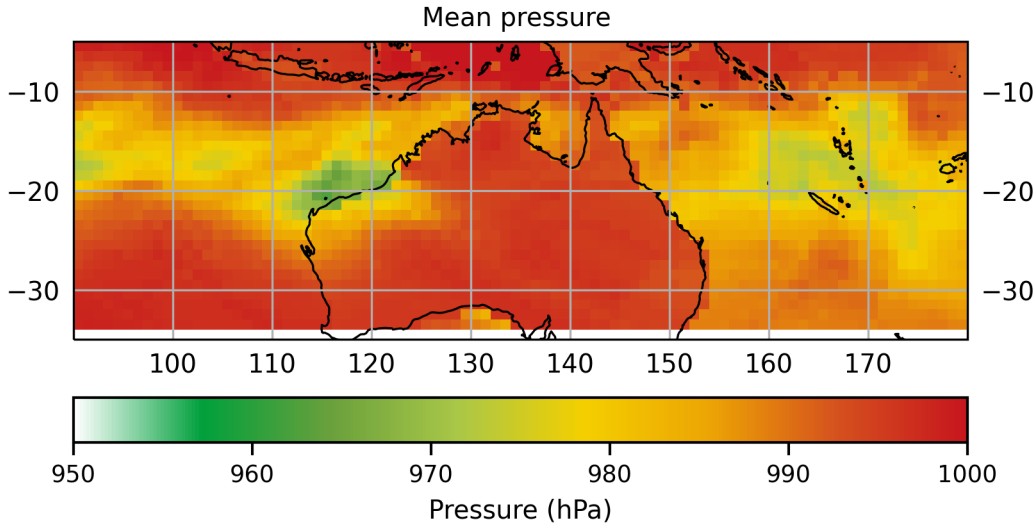

**Figure 16: Mean central pressure (hPa), based on IBTrACS v03r09 (1981-2016).**

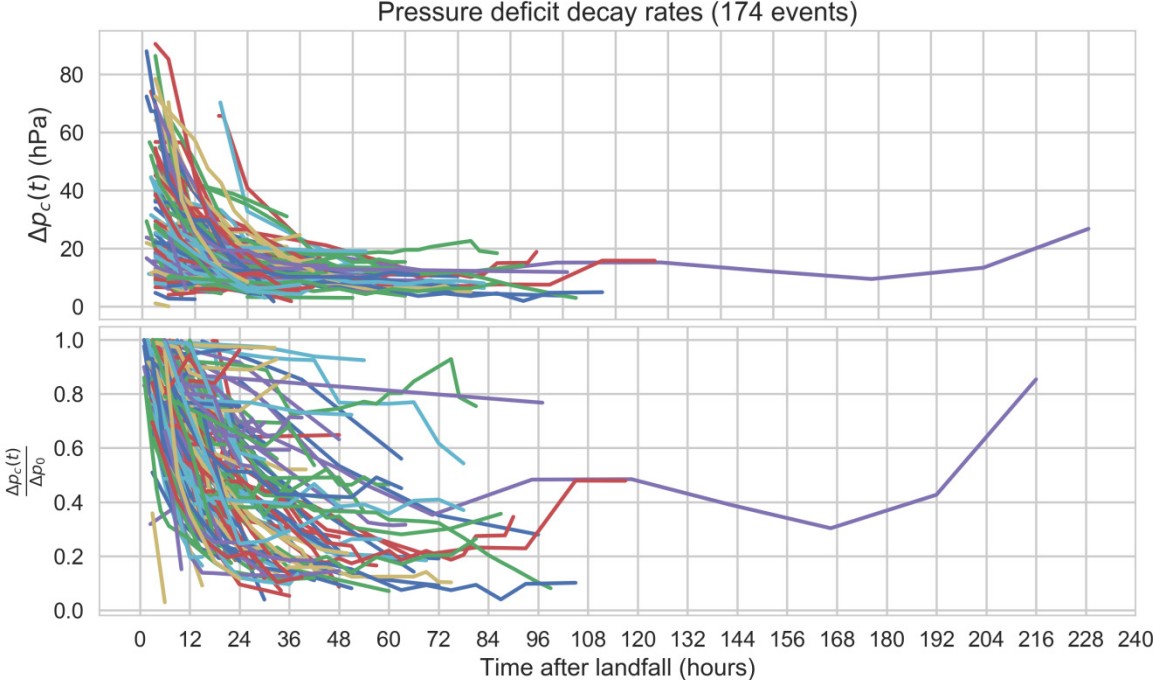

**Figure 17: Observed pressure deficit versus time after landfall (top), and normalised by pressure deficit at landfall (t=0, bottom). Colours are for clarity only.**

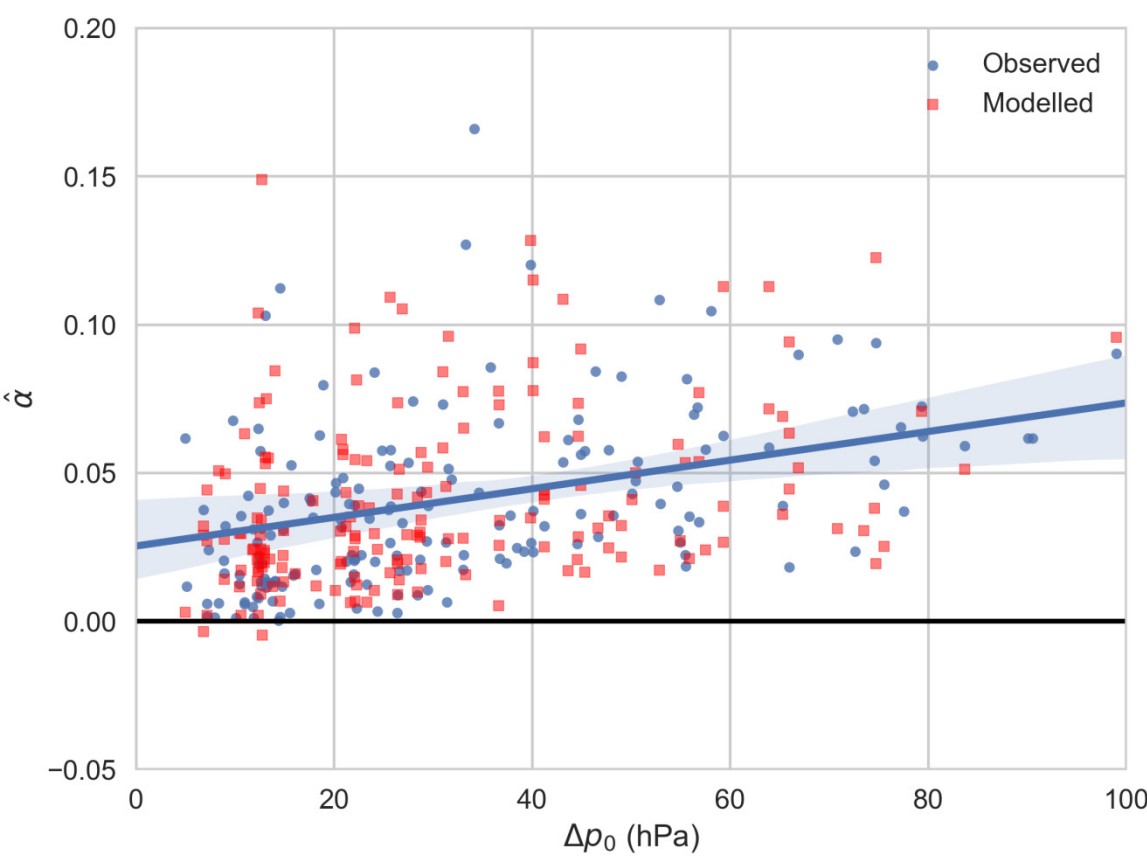

**Figure 18: Observed and modelled pressure deficit decay rates ( ) as a function of landfall pressure deficit ( ). The regression line includes the approximate 95% confidence interval (shaded) based on bootstrap resampling of the observed values. values used for the modelled decay rates are randomly sampled from the observed values. See text for details of the model equation.**

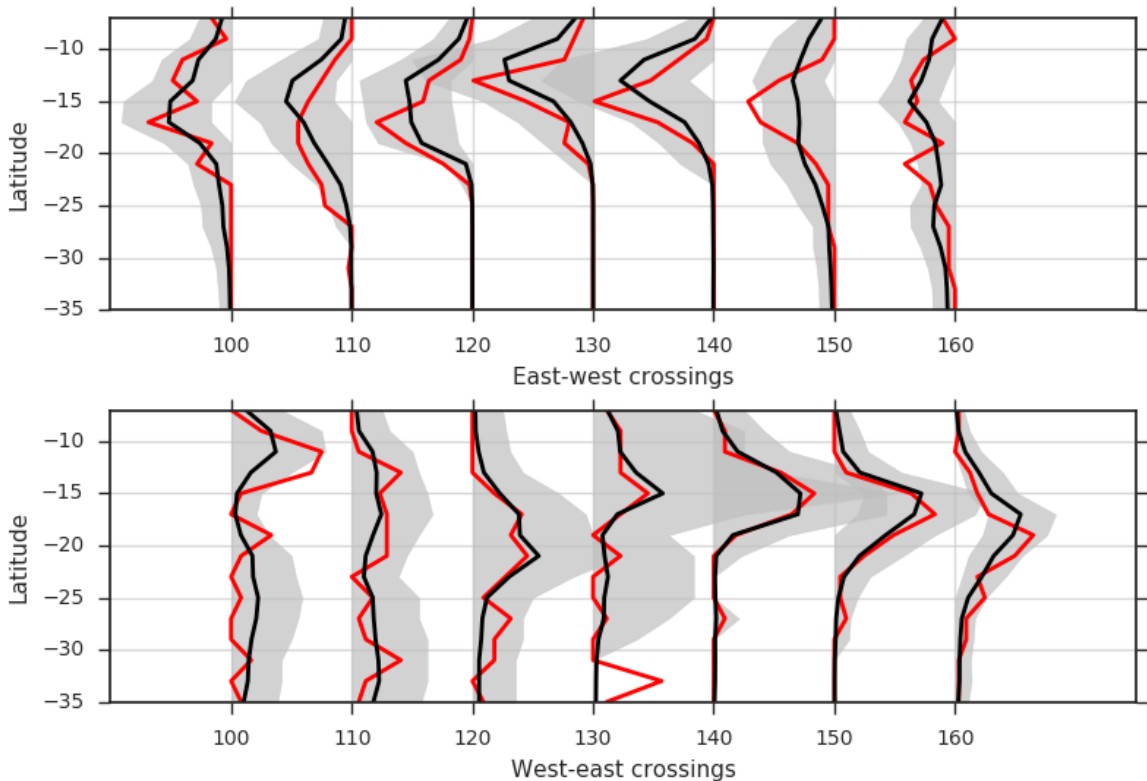

**Figure 19: Distribution of longitude crossing rates for synthetic and observed TCs. Values represent 100 times the probability density of events crossing each longitude in 2 degree latitudinal segments. Upper panel is for TCs moving east to west, lower panel for TCs moving west to east. Red lines are the observed distribution (IBTrACS v03r09 1981-2016); black the mean of 1000 simulations each representing 30 years of activity. Shaded band indicates the 90th percentile range of the simulations.**

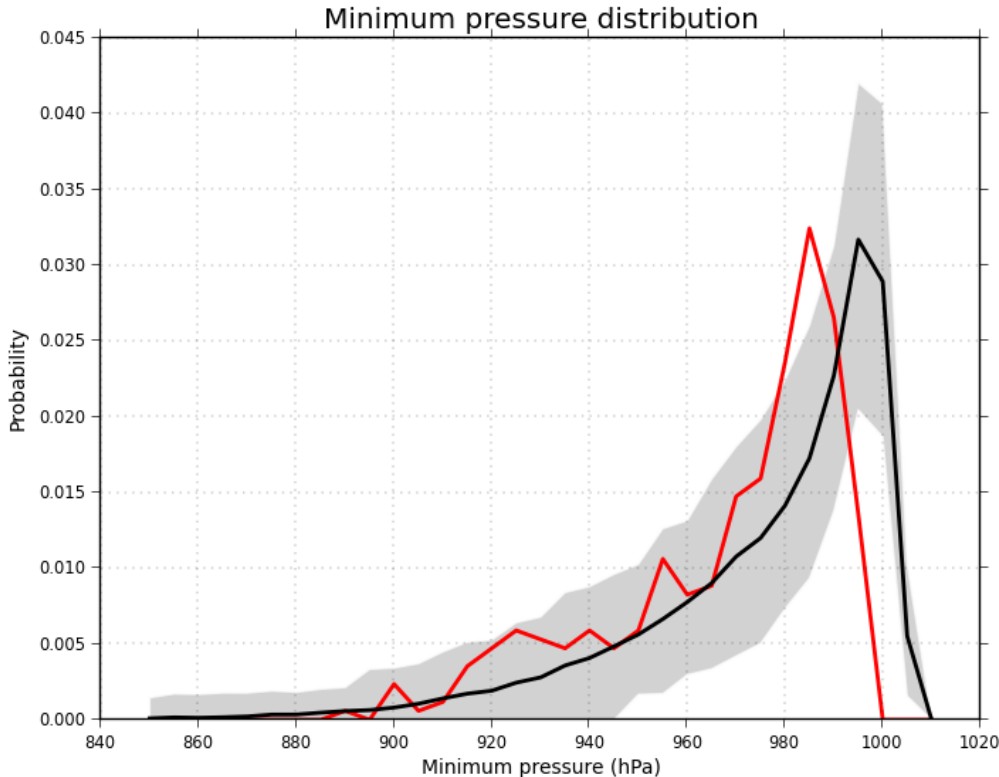

**Figure 20: Distribution of minimum central pressure values (hPa). Red line is the observed distribution (IBTrACS v03r09 1981-2016); black the mean of 1000 simulations each representing 35 years of activity. Shaded band indicates the 90th percentile range of the simulations.**

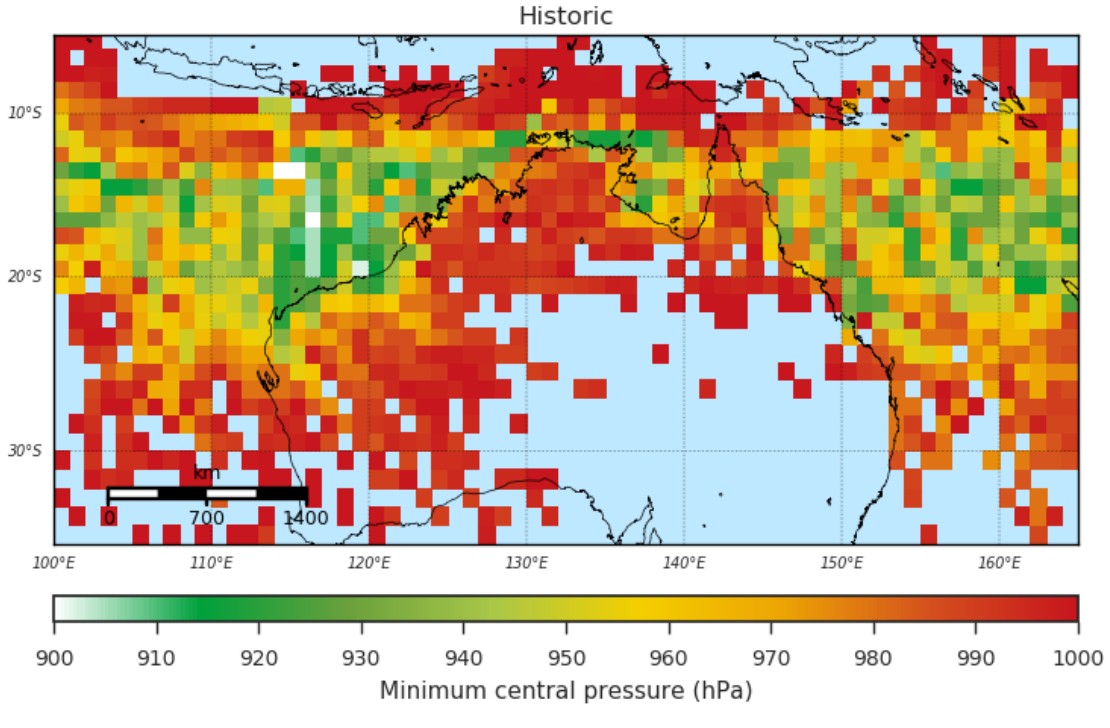

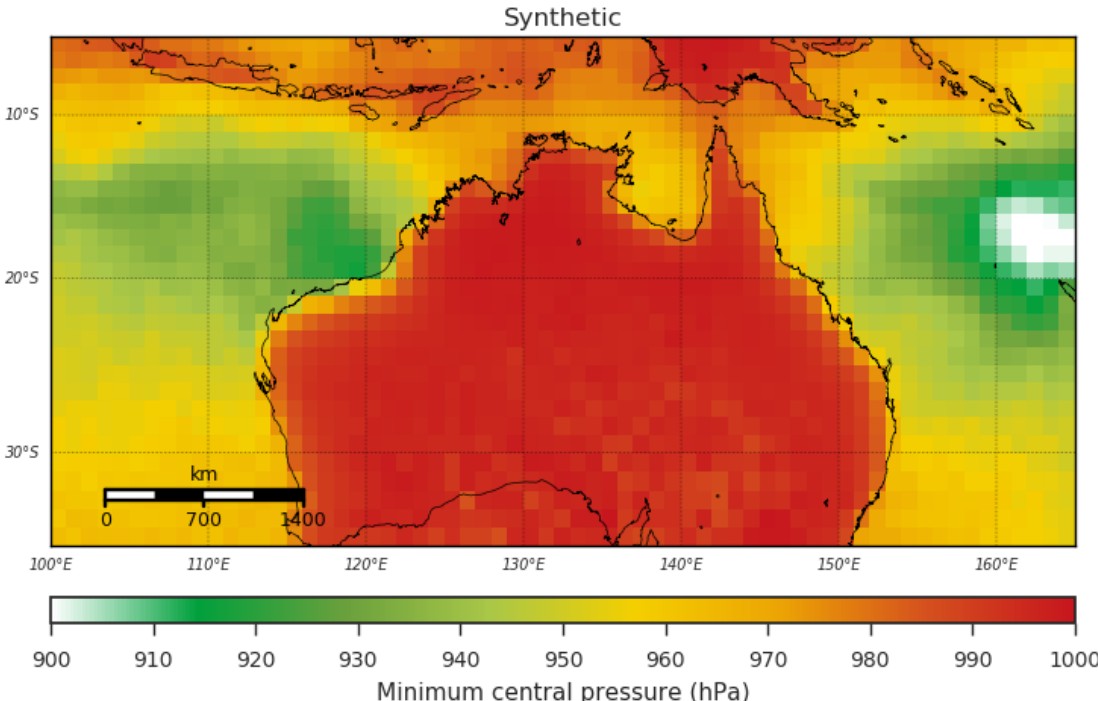

**Figure 21: Historic and synthetic minimum central pressure values over the simulation domain. Synthetic values are the mean of 1000 simulations of 35 years of TC activity.**

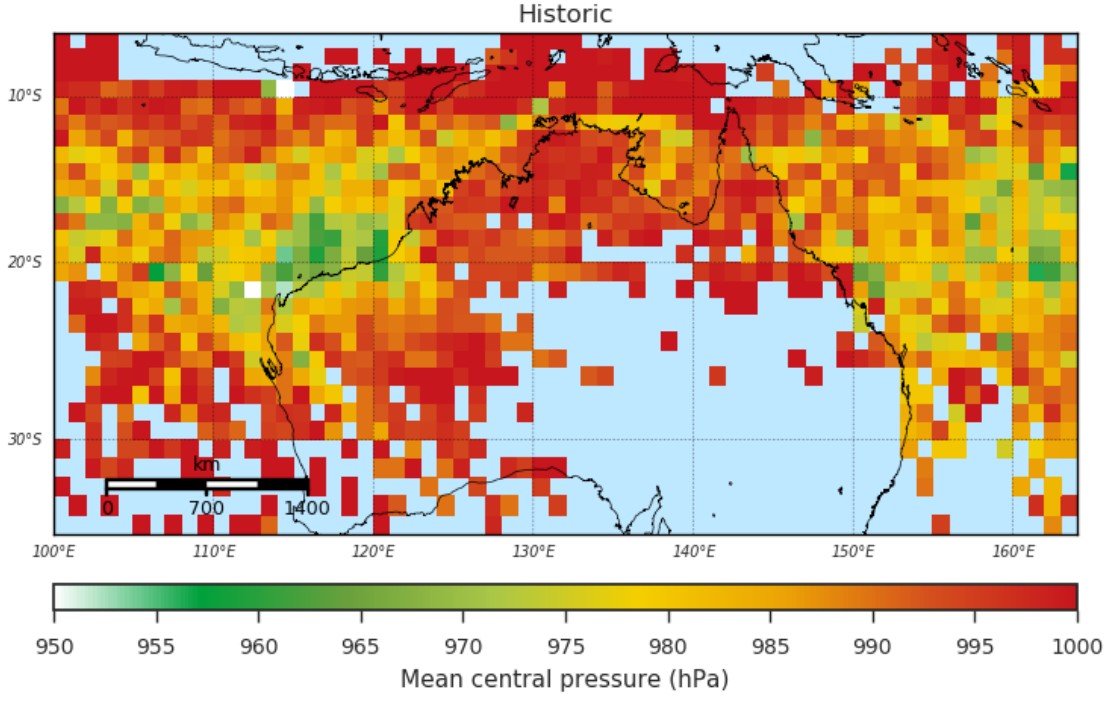

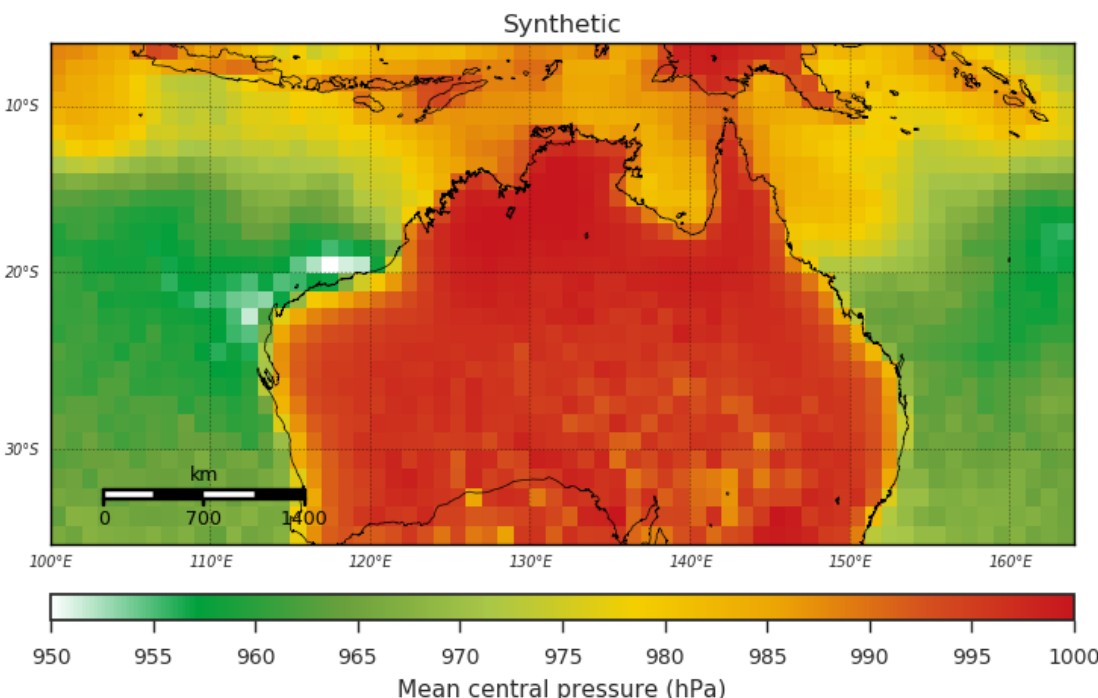

**Figure 22: Historic (top) and synthetic (bottom) mean central pressure values across the simulation domain. Synthetic values are the mean of 1000 simulations of 35 years of TC activity.**

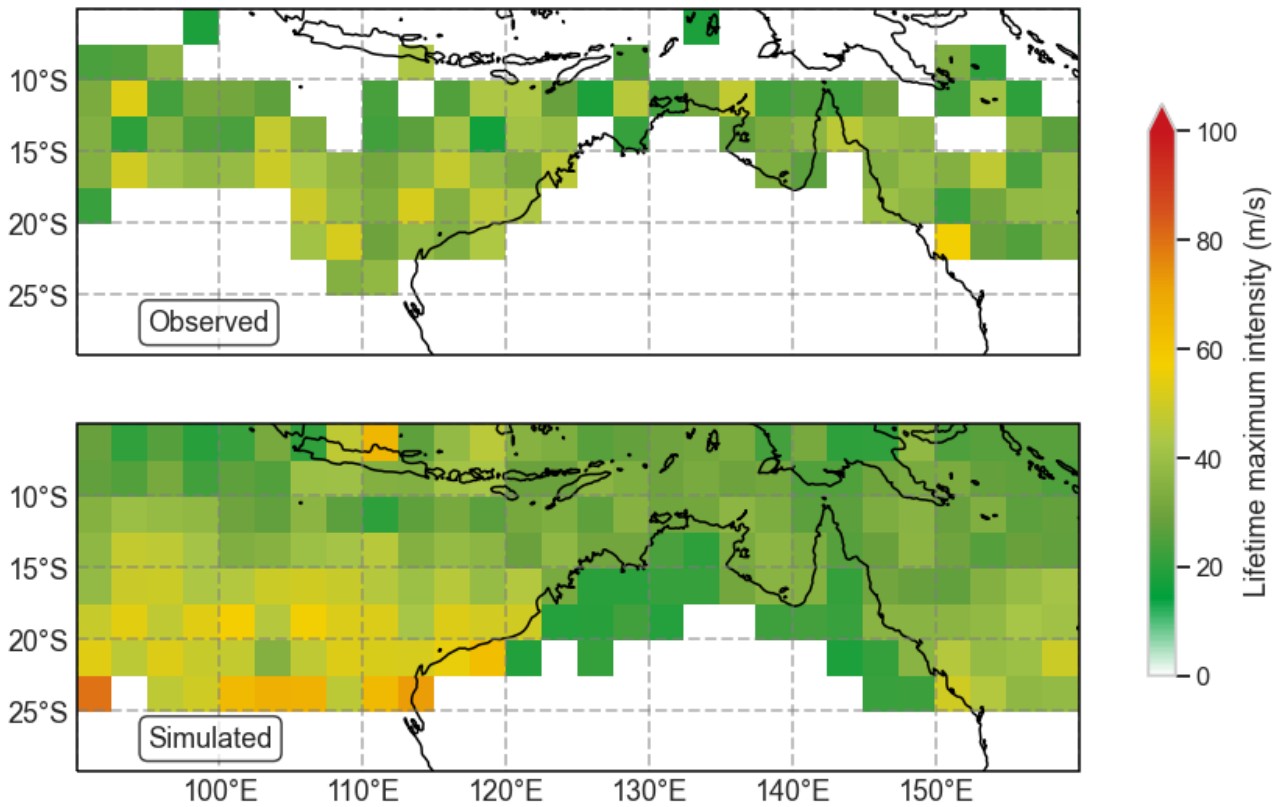

**Figure 23: Historic (top) and synthetic (bottom) mean lifetime maximum intensity. Synthetic values are the mean of 1000 simulations of 35 years of TC activity.**

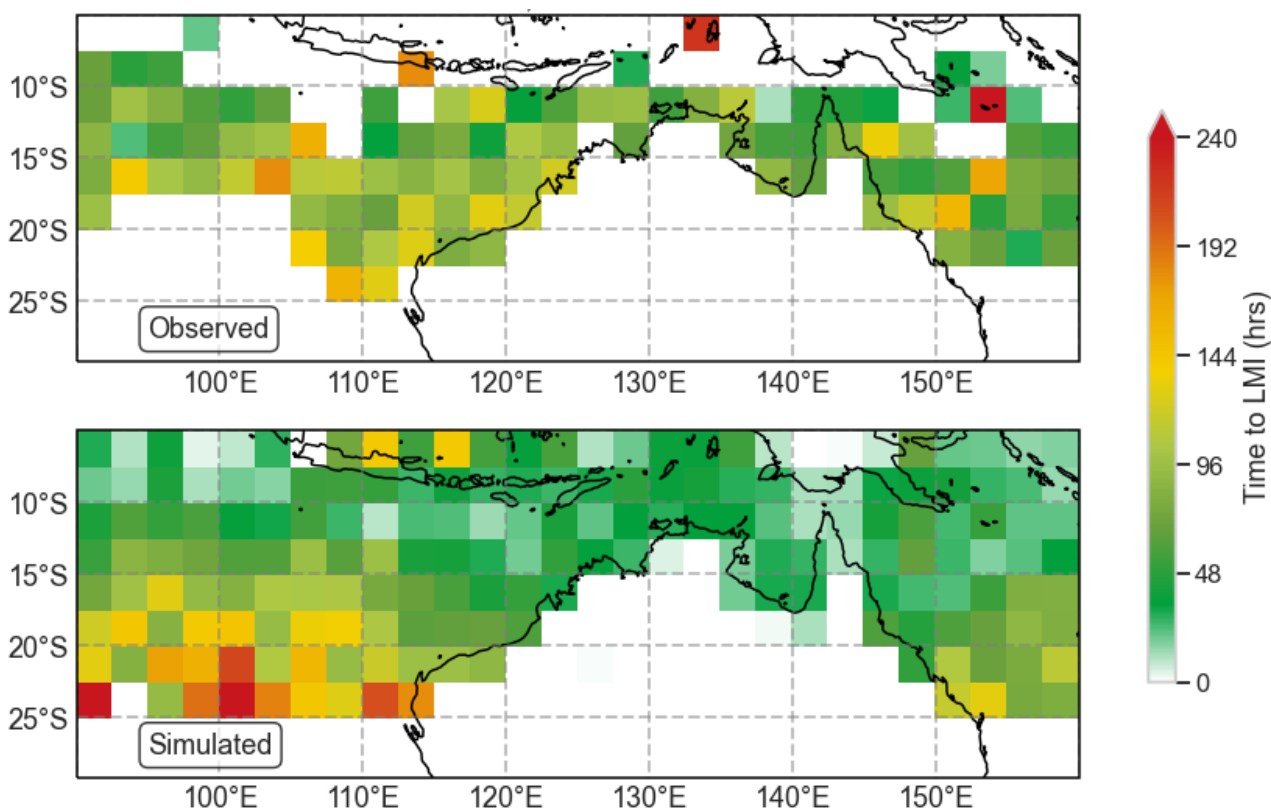

**Figure 24: Mean time taken (hours) to achieve lifetime maximum intensity for historical (top) and synthetic (bottom) TCs. Synthetic values are the mean of 1000 simulations of 35 years of TC activity.**

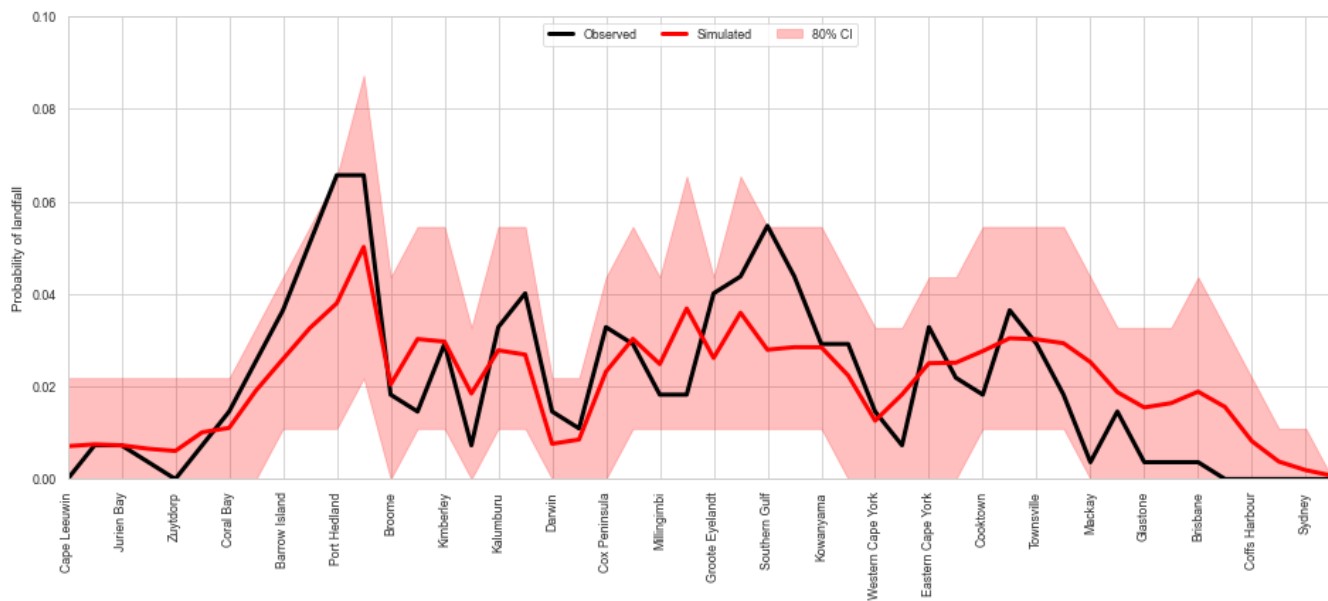

**Figure 25: Probability of a TC making landfall around the Australian coastline. Black line is the observed distribution (IBTrACS v03r09 1981-2016); red the mean of 1000 simulations each representing 35 years of activity. Shaded band indicates the 80th percentile range of the simulations.**

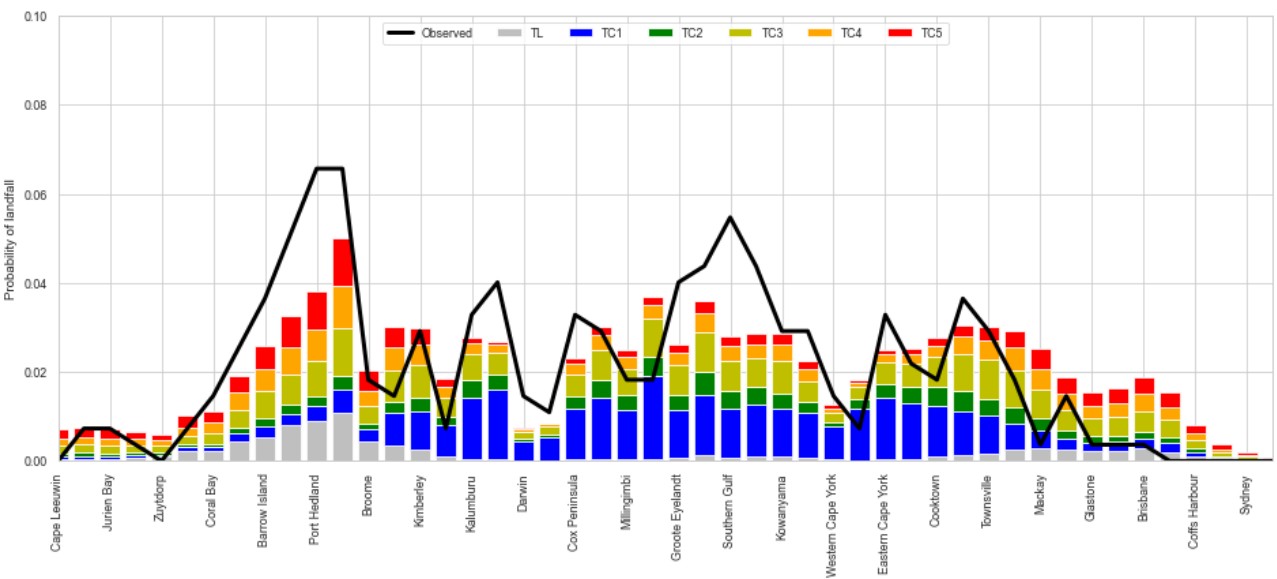

**Figure 26: Mean distribution of landfall intensity (by TC intensity category) for synthetic TCs. Categories are based on the Australian TC intensity scale. The observed landfall probability is shown in black.**

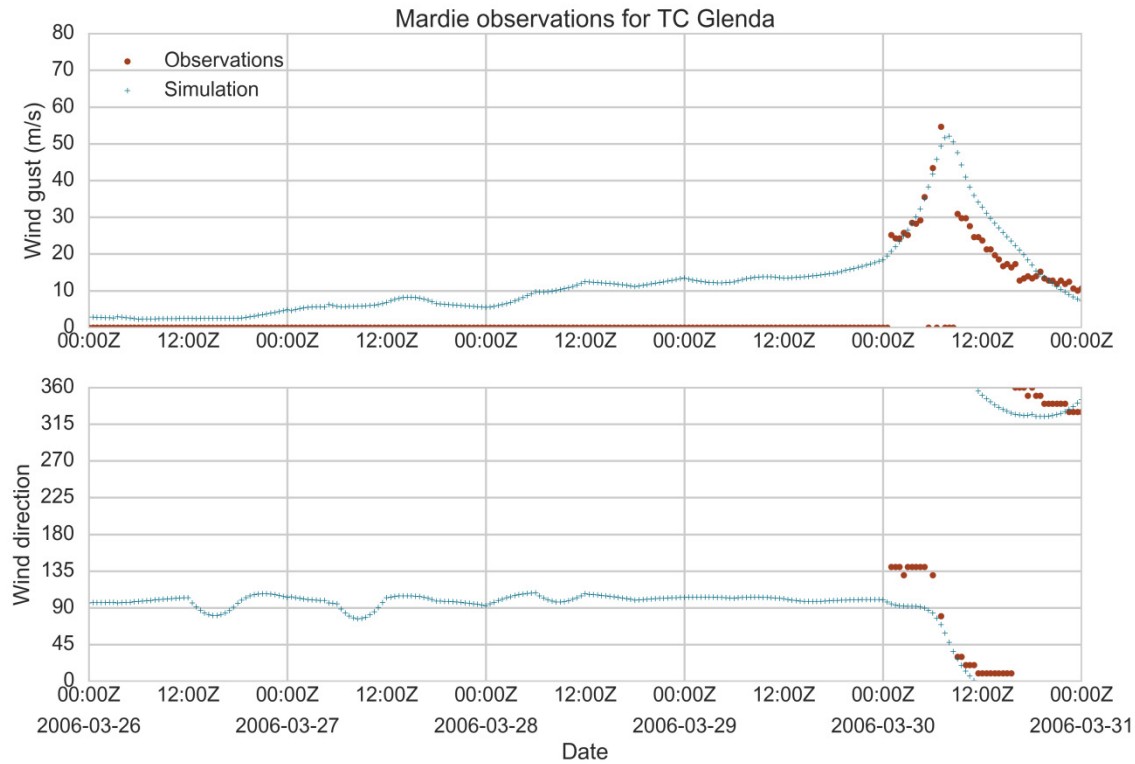

**Figure 27: Modelled and observed wind speed (top) and direction (bottom) at Mardie (Western Australia) for the passage of TC Glenda (2006).**

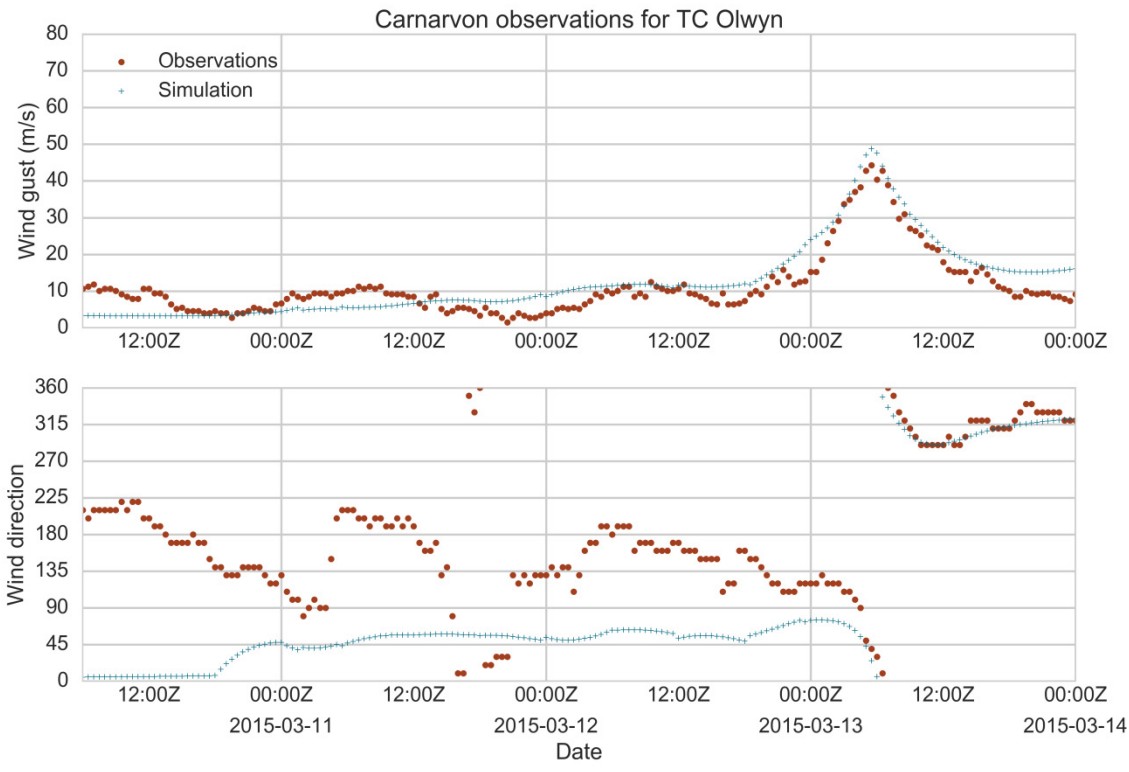

**Figure 28: Modelled and observed wind speed (top) and direction (bottom) for Carnarvon (Western Australia) for the passage of TC Olwyn (2015).**

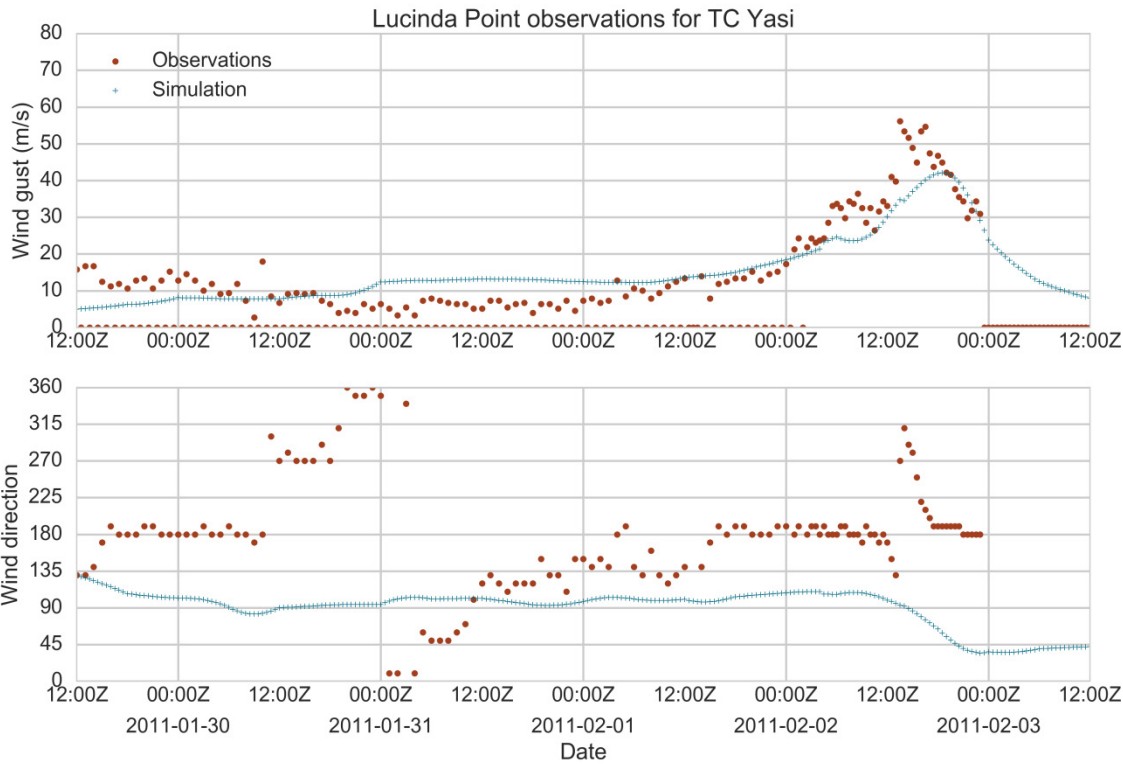

**Figure 29: Modelled and observed wind speed (top) and direction (bottom) at Lucinda Point (Queensland) for the passage of TC Yasi (2011).**

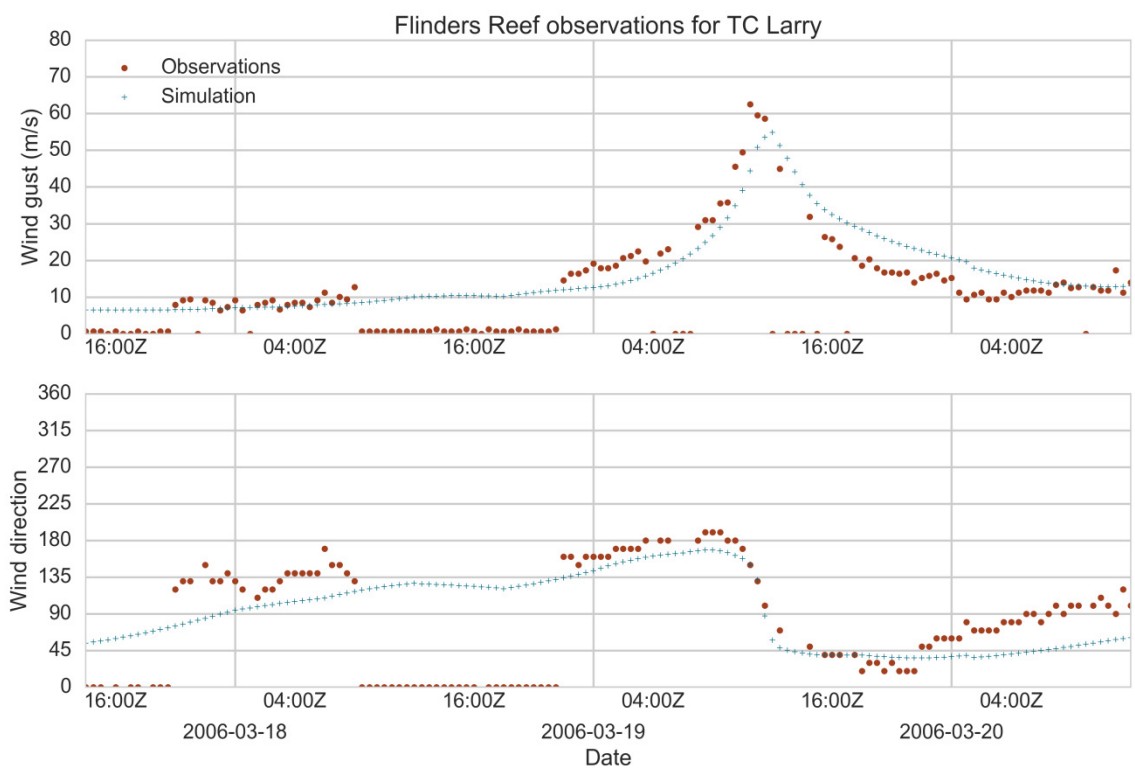

**Figure 30: Modelled and observed wind speed (top) and direction (bottom) for Flinders Reef (Queensland) for the passage of TC Larry (2006).**

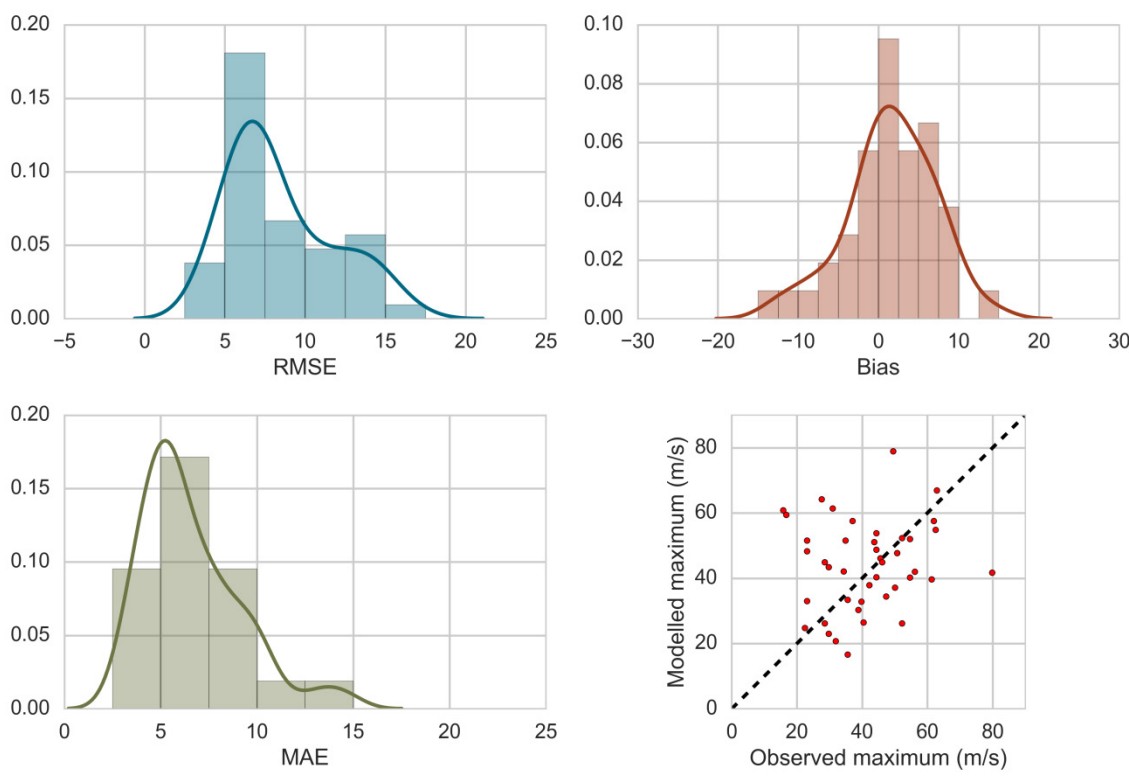

**Figure 31: Root mean square error (RMSE), bias (Bias), mean absolute error (MAE) and scatter plot of observed versus modelled maximum wind speed for 42 weather station observations associated with the passage of a TC.**

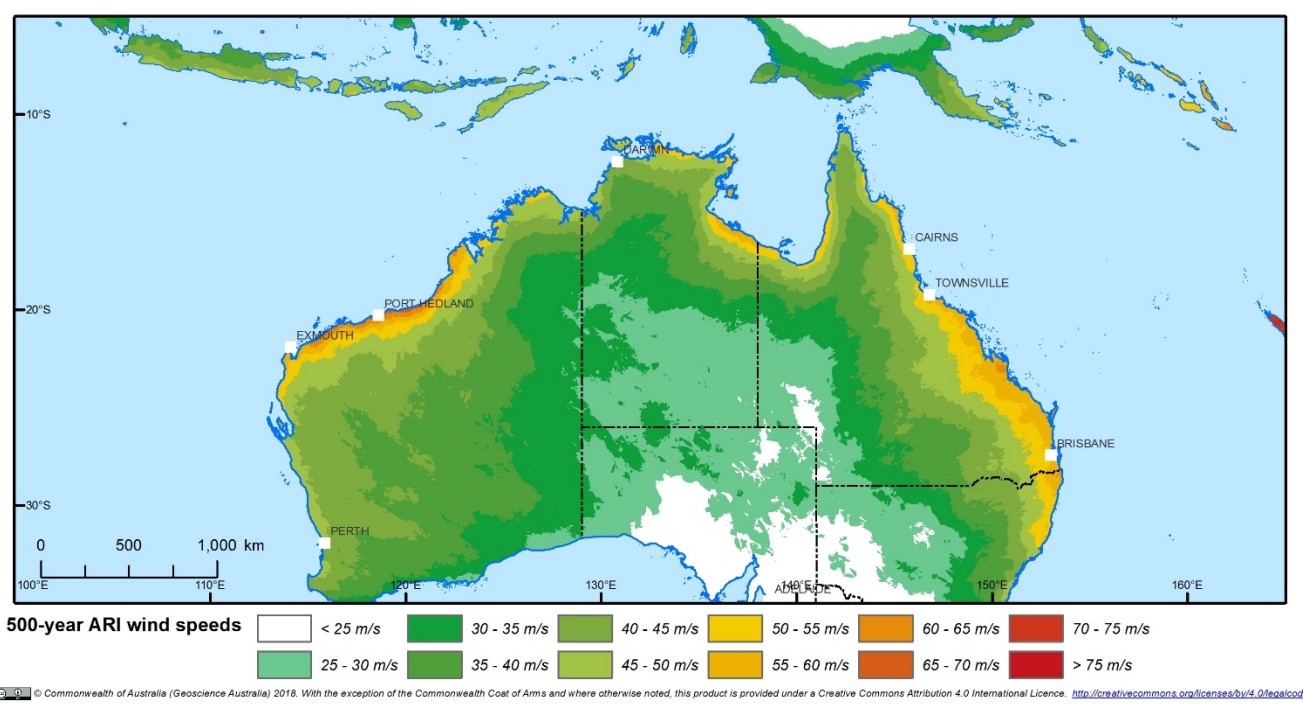

| 500-year ARI wind speeds | < 25 m/s | 30 - 35 m/s | 40 - 45 m/s | 50 - 55 m/s | 60 - 65 m/s | 70 - 75 m/s |
| 25 - 30 m/s | 35 - 40 m/s | 45 - 50 m/s | 55 - 60 m/s | 65 - 70 m/s | > 75 m/s |

**Figure 32: 500-year ARI wind speed due to tropical cyclones across Australia, using empirically estimated return period wind speeds (Eq. 11).**

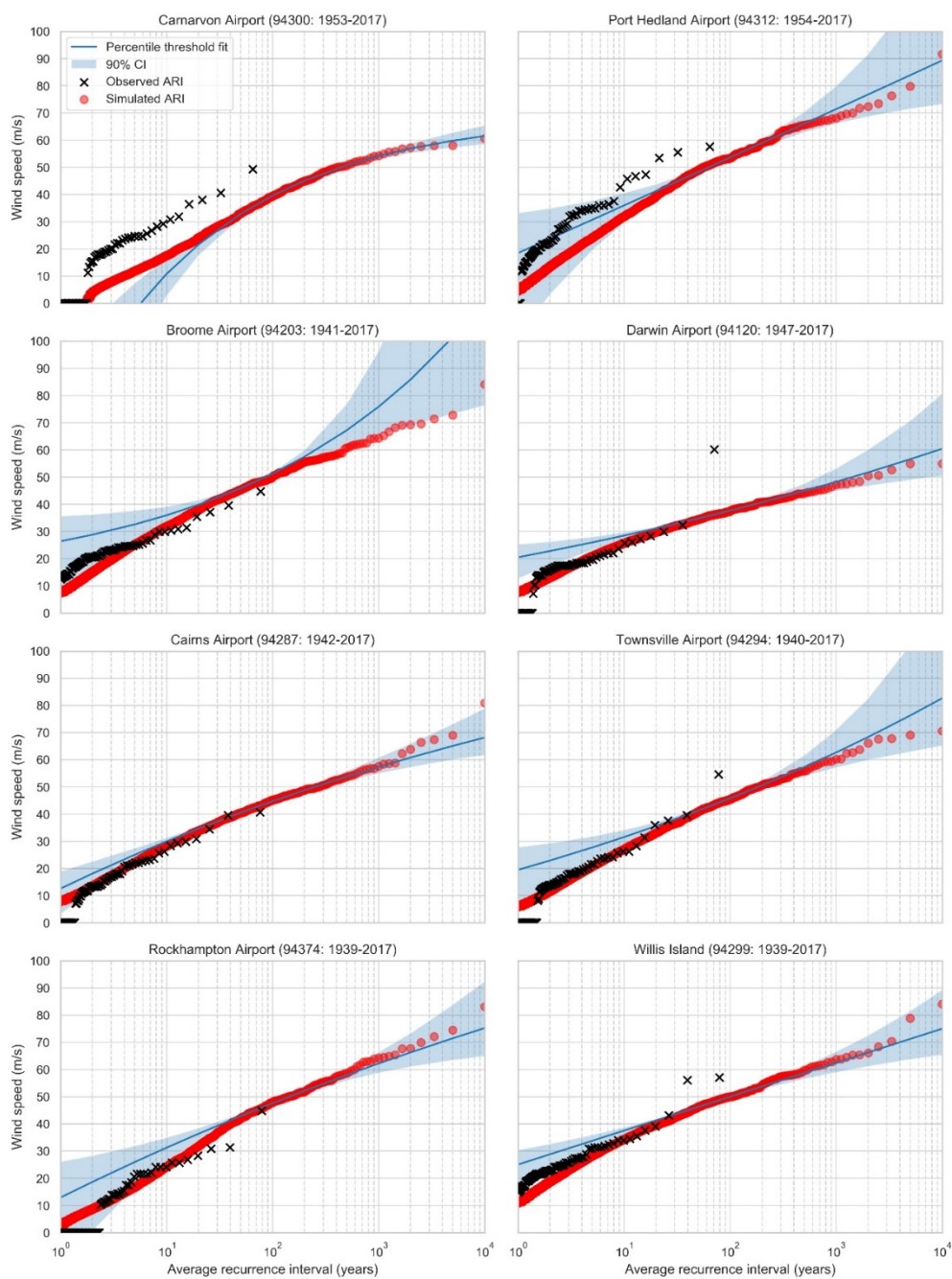

**Figure 33: Hazard profile for locations around Australia. Estimated ARI wind speeds derived from observed TC wind speeds are marked by 'X'. "Percentile threshold fit" (blue line) uses the 99.5th percentile as the threshold for the peaks-over-threshold for fitting the GPD to the simulated wind speeds from the stochastic event set (red circles). The blue shading is the 90th percentile confidence interval of the GPD fit. The title for each panel includes the years of available observational data for the location.**

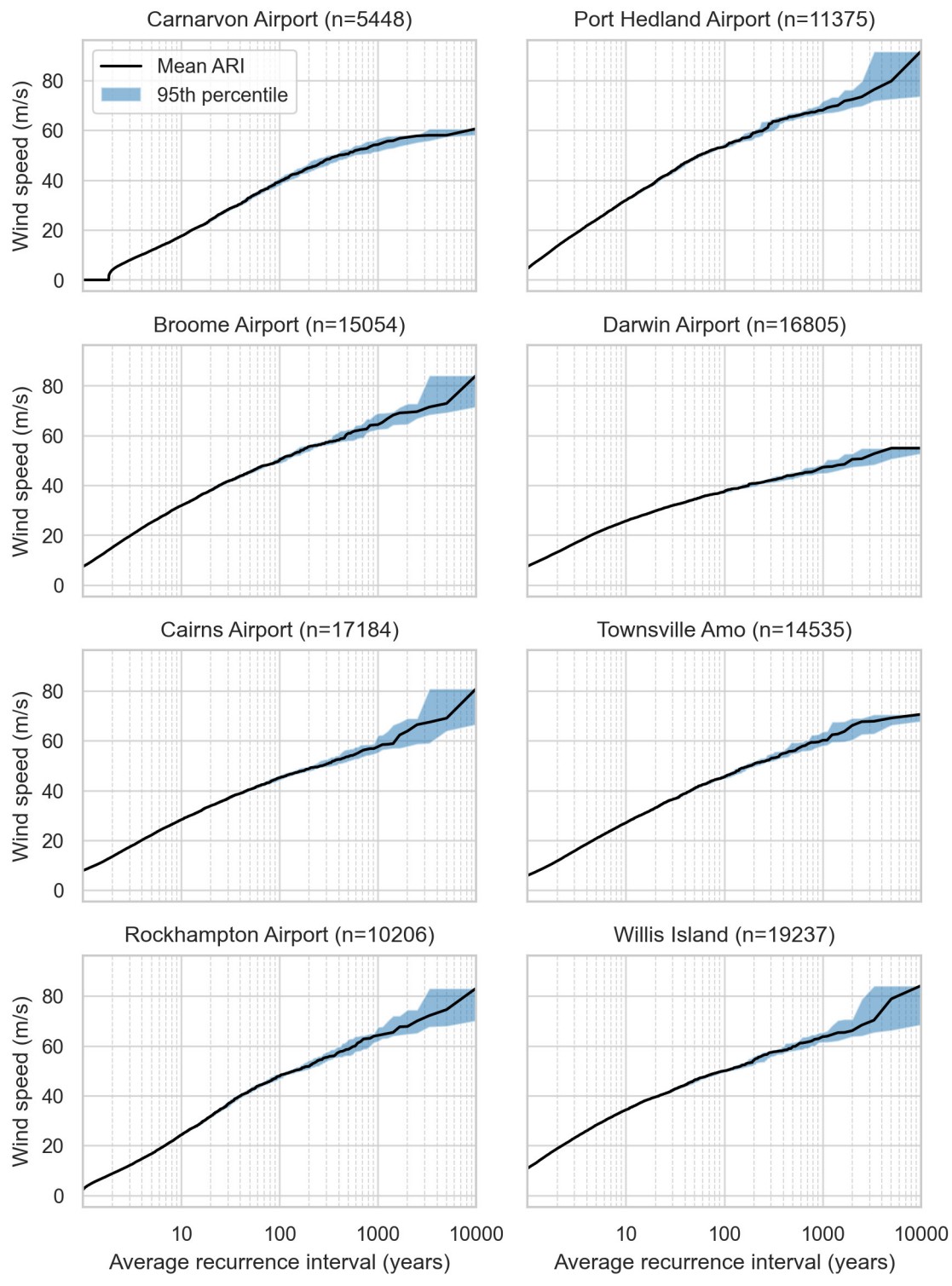

**Figure 34: ARI convergence tests for locations around Australia. The 95th percentile range is determined using bootstrap resampling.**