# Peer review of "A statistical-parametric model of tropical cyclones for hazard assessment"

_Natural Hazards and Earth System Sciences, 2019_

## Referee Comment (RC1) · James Done (Referee) · 11 Nov 2019

Summary: This study describes a model of tropical cyclone (TC) wind speed distributions that is applicable to all global TC basins. The modular approach takes collections of TC track data as input and first generates synthetic tracks for a user-defined number of years. A spatial surface wind field is then constructed along all tracks using a combination of a parametric wind profile model of the winds above the boundary layer and a boundary layer model to bring the winds down to the surface. The final stage uses extreme value theory to fill out the tail of the wind speed distribution and characterize the rare, high-impact wind speeds. An example application and evaluation of the model over Australia shows that historical TC activity is, for the most part, statistically indistinguishable from simulated TC activity.

[Figure]

I fully expect that this open source community model as presented here will have a large impact as a research tool for the broader research community and also as a risk assessment tool for the re/insurance industry and other risk managers. The paper is written with clear, logical flow and is well structured. The introduction is comprehensive in outlining the problem and motivating the work. The method is well explained, but there are a few details missing (as explained below). The subject matter is appropriate for NHESS and is well worth being published.

Specific Main Comments

1) Introduction: I understand that this model was motivated by TC risk over Australia. But this is a globally applicable model. I suggest broadening the introduction a little to also discuss global TC risk. Then focus down on Australia to motivate the case study demonstration of capability.

2) It's not clear to me the value of running the wind field model vs. simply running more synthetic years to build up enough tracks in each analysis grid cell. For example, what is the difference in the 500-year wind speed based on 100 tracks in each grid cell (no windfield module) and 100 wind field values in each grid cell (associated with tracks within and just outside each grid box)?

3) I think it's important to state more clearly the limitations of the approach in assessing TC risk. The track generator, for example, is not adding new information. It's my understanding that since it samples from the input track parameter distributions it cannot generate tracks far outside the input track distributions (unlike a free running dynamical model). Am I correct? This is important when it comes to interpreting the ARI uncertainty bounds. These uncertainty estimates are uncertainty in the model fit to the observations. These are not uncertainty bounds on the actual TC risk. The actual TC risk would need to account for uncertainty due to the short historical record. Perhaps one other limitation is that the TCRM as currently developed does not account for trends in TC frequency or TC intensity. It therefore assumes stationary statistics. It

could certainly be modified in future releases to account for temporal effects.

4) Method: Please explain why the time rate-of-change of central pressure is used rather than the absolute value of central pressure?

5) Method: There are a number of regression equations (Equations 8, 9, 12) that appear to be tuned for Australia. Are users to rederive these regression equations for their domain of interest, or are they globally applicable?

6) Method: I don't understand the need for a decay rate model (Equation 10). Isn't the decay rate already included in the input best track (pmin) data?

7) Conclusion: I think it would be useful to mention the option to additionally use local wind multiplication factors to better account for local terrain effects.

Specific Minor Comments

1) I read that it takes a few minutes to run a single scenario. Can some detail be added on the computational cost of running 1000 years?

2) Introduction: I may have missed it, but I suggest including a statement that the model can also be used for single event scenario assessments?

3) Introduction or Conclusion: I suggest adding that the model can be run with any input track data, not just historical best track data. This broadens the applications of the model to be used in conjunction with, for example, TC track data from global climate models to study climate variability and change effects on wind exceedances.

4) Section 4.5: It is stated that there are differences in the inland decay rates between the East and West coasts. But then a single decay rate model is used. Please justify this decision.

5) Section 7.1: The somewhat poor performance of the model over Northwest Australia is explained by the lower genesis probabilities. How is it possible for the model to miss these local genesis patterns if it is sampling from the genesis probability surface?

[Figure]

6) Conclusion: The introduction mentions the high cost of riverine and coastal surge flooding. Can a brief discussion be added on whether a TCRM-like approach could be used for TC rainfall and/or flooding?

7) Figure 5: Please add the units of the genesis probability.

8) Figure 13: What do the colors of the lines represent?

---

## Referee Comment (RC2) · Anonymous Referee #2 · 14 Nov 2019

Overall, this is a nice summary of a new stochastic TC model for the Australia section. The text is well written, and the Figures are clear. The methodology employed is not novel. But making the model open source is valuable and uncommon. As the model may well be used by various stakeholders, this documenting publication is definitely warranted. Before publication, however, I have some issues that need to be clarified.

1.P2L30: "L is the bandwidth matrix." This sounds as if L varies through the domain. Is that right? If so, more detail is warranted, perhaps a sentence or two stating the range and typical values of L.

2.Section 4.1. Genesis: Is there seasonality in the genesis? Or in the other model components?

[Figure]

3.P6L39-P7L1: The assumption that Vtang » Vtran is often not good even for purely tropical systems, if they're not high intensity.

4.P7L5: Additional recent relevant work on ET transition are documented in two papers by Bieli et al. (2019). The second paper, especially, addresses statistical modeling of ET transition. The references are Bieli et al, 2019, A global climatology of extratropical transition, Part I: characteristics across basins, J. Clim, 32, 3557-3582, and Bieli et al, 2019, A global climatology of extratropical transition, Part II: statistical performance of the cyclone phase space, J Clim, 32, 3583-35997)

5.P8L28-31: I don't follow this. When I look at Fig 5 I do indeed see a local maximum in the 120-130E, 10S region of the genesis probability density function. What am I missing?

6.I'm surprised that the CP model seems to work well. One of the reasons that a track model like this works is that a TC's location strongly influences its propagation, due to the role of large-scale climatological steering winds. So, it's reasonable to estimate where a TC at location X goes next by analyzing where historical TCs near X have gone next, without any additional information. But this is less true for CP. Fig 11 shows essentially two zones, north a south of roughly $15°S$. North of that, the average CP tendency is negative, and south it's positive. Many TCs in the region spend the bulk of their time in the northern negative-tendency zone. For these TCs, what keeps their CPs from declining without limit? I realize there's a large stochastic component, but, without some cap, how is the TCs intensity bounded? For example, in their stochastic TC model, James and Mason (2005) employed an elastic cap at low CP to limit decline below the CP set by the local Maximum Potential Intensity. Perhaps here, given the proximity of landmasses in the region, there's rarely an opportunity for a TC's CP to decrease below plausible meteorological limits?

Similarly, for the few TCs that form south of 15S, in the positive-tendency zone, what causes their CPs to decline on average (the storms to intensify), as opposed to just

quickly attenuating to lysis? Is some kind of filter applied, to only accept storms that stochastically intensify beyond some threshold?

Discussion about these points is warranted. In addition, it would be helpful to show some examples of over-ocean CP time series from stochastic simulations. It would also be helpful to have additional panels in Fig 19 that show landfall probabilities for TCs with CP below specified thresholds. (I assume the current version of Fig 19 applies to all TCs, regardless of CP.) Finally, I'd like to see the magnitude of the CP sigma, perhaps in a second panel of Fig 11, to compare to the mean tendencies.

7.Figs 1 and 19: It would help to have mileposts (e.g., towns) indicated on the landfall profile of Fig 19 and correspondingly on the map of Australia, perhaps the ma of Fig 1.

---

## Author Comment (AC1) · 24 Jul 2020

1) Introduction: I understand that this model was motivated by TC risk over Australia. But this is a globally applicable model. I suggest broadening the introduction a little to also discuss global TC risk. Then focus down on Australia to motivate the case study demonstration of capability.

There is commentary in the introduction on the capability to apply the model in other basins. However, the author has not preformed any in-depth tests in other basins. We are aware of applications in other basins (unpublished).

2) It's not clear to me the value of running the wind field model vs. simply running more synthetic years to build up enough tracks in each analysis grid cell. For example, what

is the difference in the 500-year wind speed based on 100 tracks in each grid cell (no windfield module) and 100 wind field values in each grid cell (associated with tracks within and just outside each grid box)?

The zone of strongest winds is displaced to the left of the track (in the southern hemisphere). Using vmax at the track location will give a different result to simulating the full wind field for a given grid cell where the maximum winds will be simulated several grid cells (using a 0.02-degree grid cell) away from the track. Given the desired resolution of the resulting ARI wind speed data, it is more efficient to calculate wind fields. This also offers the opportunity to deliver wind field data to end users from individual events.

3) I think it's important to state more clearly the limitations of the approach in assessing TC risk. The track generator, for example, is not adding new information. It's my understanding that since it samples from the input track parameter distributions it cannot generate tracks far outside the input track distributions (unlike a free running dynamical model). Am I correct? This is important when it comes to interpreting the ARI uncertainty bounds. These uncertainty estimates are uncertainty in the model fit to the observations. These are not uncertainty bounds on the actual TC risk. The actual TC risk would need to account for uncertainty due to the short historical record. Perhaps one other limitation is that the TCRM as currently developed does not account for trends in TC frequency or TC intensity. It therefore assumes stationary statistics. It could certainly be modified in future releases to account for temporal effects.

I agree on the leading point here - we are restricted by the input track distributions. The uncertainty in ARI wind speeds beyond, say 100 years, using point observations is becomes very large, as the observations are often less than 50 years (in the Australian region at least). Using this track model produces a larger synthetic record that is in line with the observed record, but also extends it. For example, teh model allows intensity to exceed the observed record of intensity. We note that the short term record (35 years) is a restriction on the actual TC hazard. It does assume stationary statistics, and further validation against statistics such as palaeotempestology to place the ARI

wind speeds in a broader context should be considered. Our current intent in this paper is to describe the model and initial validation exercises. The manuscript has been updated to add discussion on uncertainty arising from the short input record, and how this should be considered in longer time scale contexts.

4) Please explain why the time rate-of-change of central pressure is used rather than the absolute value of central pressure?

Pressure tendency is preferred to absolute pressure due to the lower lag-1 autocorrelation in the tendency values. Using absolute values leads to rapid and almost one-way variation (i.e. constant increase or decrease) in the intensity. There remains a strong autocorrelation beyond lag-1 for absolute pressure values, but not for pressure tendency values. Additional figures have been included in the manuscript for autocorrelation of variables (Fig 1).

5) Method: There are a number of regression equations (Equations 8, 9, 12) that appear to be tuned for Australia. Are users to rederive these regression equations for their domain of interest, or are they globally applicable?

Regression equations are based on Australian data, in line with the choice of case study. As the model is free and open source, it is possible for users to modify these regressions for their chosen domain. Supporting software demonstrates how the regression equations were derived, and can be used to determine those regressions for other regions. The manuscript has been updated to expand discussion on the formulation of the regression equations, and highlight these are regionally specific but can be adapted to other regions. Noting the intention is that the model is not intended to be run on a global domain, but on basin-wide domains similar to other track modelling systems (e.g. Hall & Jewson 2007).

6) Method: I don't understand the need for a decay rate model (Equation 10). Isn't the decay rate already included in the input best track (pmin) data?

Using the autoregressive model over land led to an abundance of storms that reintensified, and generated many more unrealistically long-lived tracks. Implementing the decay rate model forces the TCs to decay in line with observed events (but does allow for very occasional over-land intensification).

7) I think it would be useful to mention the option to additionally use local wind multiplication factors to better account for local terrain effects.

Noted. Updated manuscript to include reference to the Krause & Arthur (2018) that describes the approach in detail.

Minor comments: 1) I read that it takes a few minutes to run a single scenario. Can some detail be added on the computational cost of running 1000 years?

The windfield for a single scenario takes about 10 minutes to run, depending on the overall extent of the track. Using multiprocessor systems, a hazard simulation, including track generation and wind fields across the Australian region (100-170E, 5-35S) for a 10,000 year catalogue requires around 3000 hours of CPU time.

2) I may have missed it, but I suggest including a statement that the model can also be used for single event scenario assessments? The model can and is regularly used for single scenario simulation. At GA, we use the model to provide guidance on wind field zones for Emergency Management Authorities based on track information provided by the Bureau of Meteorology (BoM). We can provide the wind fields to EMs on average within 30 minutes of publication of track forecasts from the BoM. See paragraph 5 of the Introduction.

3) Introduction or Conclusion: I suggest adding that the model can be run with any input track data, not just historical best track data. This broadens the applications of the model to be used in conjunction with, for example, TC track data from global climate models to study climate variability and change effects on wind exceedances.

Noted - we are presently working on using other input sources to demonstrate this

capability. Manuscript updated to note other data sources can be used.

4) Section 4.5: It is stated that there are differences in the inland decay rates between the East and West coasts. But then a single decay rate model is used. Please justify this decision.

The driver of different decay rates in the east and west is topography. To minimise the data demands (especially with a view to global application), we did not include topography in the regression such that we do not have to source suitable topographic data for all potential basins. This is an area for future development.

5) Section 7.1: The somewhat poor performance of the model over Northwest Australia is explained by the lower genesis probabilities. How is it possible for the model to miss these local genesis patterns if it is sampling from the genesis probability surface?

I am yet to fully explore the causes of this issue in the reported version of the model. Other users have anecdotally noted relatively large number of short-lived weak systems in some basins, and we are now investigating methods to resolve this issue.

6) Conclusion: The introduction mentions the high cost of riverine and coastal surge flooding. Can a brief discussion be added on whether a TCRM-like approach could be used for TC rainfall and/or flooding?

Noted. Manuscript updated with a comment on application to other hazards - for example inclusion of a parameterised rainfall model for rain rates, use as input in surge modelling and wave modelling applications.

7) Figure 5: Please add the units of the genesis probability

Caption updated.

8) Figure 13: What do the colors of the lines represent?

Colours are for clarity only. Noted in caption.

[Figure]

[Figure]

[Figure]

**Fig. 1.** Autocorrelation of minimum central pressure (left) and pressure rate of change (right) for lagged observations between 1 and 20 steps. See manuscript for more details.

**Fig. 2.** TC genesis points for historical TC events (1981-2016), and the corresponding probability density function (TCs/year).

---

## Author Comment (AC2) · 24 Jul 2020

1) P2L30: "L is the bandwidth matrix." This sounds as if L varies through the domain. Is that right? If so, more detail is warranted, perhaps a sentence or two stating the range and typical values of L.

The bandwidth matrix is a 2x1 array that defines the bandwidth for x- and y-variation of the kernel function. Updated manuscript to better describe the bandwidth matrix.

2) Section 4.1. Genesis: Is there seasonality in the genesis? Or in the other model components?

Seasonality is considered in the genesis of events and the environmental pressure, but is not considered in other model components. To do so would detrimentally af-

fect the sample sizes for the gridded statistical values applied in the track generation component.

3) P6L39-P7L1: The assumption that Vtang Âż Vtran is often not good even for purely tropical systems, if they're not high intensity.

We have also noted this, and remain cautious of the resulting implications for ARI wind speeds at low intensity. Commentary added to the manuscript in section 5.2

4) P7L5: Additional recent relevant work on ET transition are documented in two papers by Bieli et al. (2019). The second paper, especially, addresses statistical modeling of ET transition. The references are Bieli et al, 2019, A global climatology of extratropical transition, Part I: characteristics across basins, J. Clim, 32, 3557-3582, and Bieli et al, 2019, A global climatology of extratropical transition, Part II: statistical performance of the cyclone phase space, J Clim, 32, 3583-35997)

Noted. Most relevant is the pathways to ET, and the likelihood of each in different basins and the implications for parametric wind fields.

5) P8L28-31: I don't follow this. When I look at Fig 5 I do indeed see a local maximum in the 120-130E, 10S region of the genesis probability density function. What am I missing?

The genesis probability for simulations is not shown, only the observed genesis probability.

6) I'm surprised that the CP model seems to work well. One of the reasons that a track model like this works is that a TC's location strongly influences its propagation, due to the role of large-scale climatological steering winds. So, it's reasonable to estimate where a TC at location X goes next by analyzing where historical TCs near X have gone next, without any additional information. But this is less true for CP. Fig 11 shows essentially two zones, north a south of roughly 15S. North of that, the average CP tendency is negative, and south it's positive. Many TCs in the region spend the bulk

of their time in the northern negative-tendency zone. For these TCs, what keeps their CPs from declining without limit? I realize there's a large stochastic component, but, without some cap, how is the TCs intensity bounded? For example, in their stochastic TC model, James and Mason (2005) employed an elastic cap at low CP to limit decline below the CP set by the local Maximum Potential Intensity. Perhaps here, given the proximity of landmasses in the region, there's rarely an opportunity for a TC's CP to decrease below plausible meteorological limits?

To minimise the volume of ancillary data that is needed to support the model initial development we aimed for a purely statistical approach. The lower limit for the pressure deficit is set to $\mu + 5^*\sigma$ for the grid cell. However, we note that PI is potentially a more instructive limit, and we are presently working on enhancements that will consider this. Additional commentary added to section 4.2.

7) Similarly, for the few TCs that form south of 15S, in the positive-tendency zone, what causes their CPs to decline on average (the storms to intensify), as opposed to just quickly attenuating to lysis? Is some kind of filter applied, to only accept storms that stochastically intensify beyond some threshold?

Storms that do not intensify beyond a pressure deficit of 5 hPa, or survive for more than 12 hours are discarded. Additional commentary in section 4.1.

8) Discussion about these points is warranted. In addition, it would be helpful to show some examples of over-ocean CP time series from stochastic simulations. It would also be helpful to have additional panels in Fig 19 that show landfall probabilities for TCs with CP below specified thresholds. (I assume the current version of Fig 19 applies to all TCs, regardless of CP.) Finally, I'd like to see the magnitude of the CP sigma, perhaps in a second panel of Fig 11, to compare to the mean tendencies.

Added figures of time to maximum lifetime intensity for observed and simulated events (Fig 21 in the manuscript). Added figure of CP sigma (Figure 12 in the manuscript). Added figure of categorised TC landfall (Figure 23).

9) Figs 1 and 19: It would help to have mileposts (e.g., towns) indicated on the landfall profile of Fig 19 and correspondingly on the map of Australia, perhaps the map of Fig 1.

Noted. Updated figure accordingly.

———————————————

[Figure]

Mean pressure rate

Pressure rate (hPa/hr)

[Figure]

Pressure rate standard deviation

$\sigma$ (hPa/hr)

[Figure]

**Fig. 1.** Mean (top) and standard deviation (bottom) of rate of change of central pressure (hPa/hour), based on IBTrACS v03r09 (1981-2016).

[Figure]

**Fig. 2.** Updated Figure 1 including milepoosts around the coastline

[Figure]

**Fig. 3.** Historic (top) and synthetic (bottom) mean lifetime maximum intensity. Synthetic values are the mean of 1000 simulations of 35 years of TC activity.

**Fig. 4.** Mean time taken (hours) to achieve lifetime maximum intensity for historical (top) and synthetic (bottom) TCs. Synthetic values are the mean of 1000 simulations of 35 years of TC activity.

**Fig. 5.** Mean distribution of landfall intensity (by TC intensity category) for synthetic TCs. Categories are based on the Australian TC intensity scale. The observed landfall probability is shown in black.

---

## Author Response (AR1)

| Reviewer 1 comments | Response comments | Changes |
|---|---|---|
| I understand that this model was motivated by TC risk over Australia. But this is a globally applicable model. I suggest broadening the introduction a little to also discuss global TC risk. Then focus down on Australia to motivate the case study demonstration of capability. | | |
| It's not clear to me the value of running the wind field model vs. simply running more synthetic years to build up enough tracks in each analysis grid cell. For example, what is the difference in the 500-year wind speed based on 100 tracks in each grid cell (no windfield module) and 100 wind field values in each grid cell (associated with tracks within and just outside each grid box)? | Firstly, we use the pressure deficit as the modelled intensity parameter in the track generation module. The wind fields are calculated in a subsequent step. Secondly, the zone of strongest winds is displaced to the left of the track (in the southern hemisphere). Using vmax at the track location will give a different result to simulating the full wind field for a given grid cell where the maximum winds will be simulated several grid cells (using a 0.02-degree grid cell) away from the track. Given the desired resolution of the resulting ARI wind speed data, it is more efficient to calculate wind fields. This approach also offers the opportunity to deliver wind field data to end users from individual events. | Added text at Section 5: Parametric wind fields are calculated for each event in the synthetic catalogue to enable a high spatial resolution understanding of the ARI wind speeds. The additional benefit of this calculation is that users can select individual synthetic events from the catalogue and obtain a wind field for use in scenario simulations. |

| | | |
|---|---|---|
| I think it's important to state more clearly the limitations of the approach in assessing TC risk. The track generator, for example, is not adding new information. It's my understanding that since it samples from the input track parameter distributions it cannot generate tracks far outside the input track distributions (unlike a free running dynamical model). Am I correct? This is important when it comes to interpreting the ARI uncertainty bounds. These uncertainty estimates are uncertainty in the model fit to the observations. These are not uncertainty bounds on the actual TC risk. The actual TC risk would need to account for uncertainty due to the short historical record. Perhaps one other limitation is that the TCRM as currently developed does not account for trends in TC frequency or TC intensity. It therefore assumes stationary statistics. It could certainly be modified in future releases to account for temporal effects. | I agree on the leading point here - we are restricted by the input track distributions. The uncertainty in ARI wind speeds beyond, say 100 years, using point observations is becomes very large, as the observations are often less than 50 years (in the Australian region at least). Using this track model produces a larger synthetic record that is in line with the observed record, but also extends it. For example, the model allows intensity to exceed the observed record of intensity.

We note that the short term record (35 years) is a restriction on the actual TC hazard. It does assume stationary statistics, and further validation against statistics such as palaeotempestology to place the ARI wind speeds in a broader context should be considered. Our current intent in this paper is to describe the model and initial validation exercises. | Added the following to section 2:
Further, the absolute accuracy of the input data is viewed as a source of uncertainty in the hazard values presented here. For example, Courtney and Burton (2019) reported on progress to improve the best track archive in Australia, noting the reassessment of intensity due to improved reanalysis methods. Such changes in the intensity values will flow through the hazard model to produce changes in the likelihood of extreme wind speeds. A thorough treatment of the accuracies arising from changes in the best track is warranted (Harper et al., 2008), and the hazard values herein should be considered as only one view of the true wind hazard arising from TC events. Yet another aspect that remains to be explored is the effect of centennial and longer variability in TC activity (Haig et al., 2014; Nott et al., 2007). |
| Please explain why the time rate-of-change of central pressure is used rather than the absolute value of central pressure? | Pressure tendency is preferred to absolute pressure due to the lower lag-1 autocorrelation in the tendency values. Using absolute values leads to rapid and almost one-way variation (i.e. constant increase or decrease) in the intensity. There remains a strong autocorrelation beyond lag-1 for absolute pressure values, but not for pressure tendency values. | Added plots of the autocorrelation for pressure and pressure rate of change to section 4.2 (pg 5.) as well as the following text: $\dot{p}$ is preferred to absolute pressure deficit due to the lower lag-1 autocorrelation in the tendency values, making it more akin to a true Markov process than simulating absolute pressure deficit. Figure 6 shows the autocorrelation for both $\dot{p}$ and $p$ for a selected grid cell in the Coral Sea. In this case, the lag-1 autocorrelation of $\dot{p}$ is 0.3, compared to that of $p$ which is 0.79. Using absolute values leads to rapid and almost one-way variation (i.e. constant increase or decrease) in the intensity. There remains a strong autocorrelation beyond |

| | | lag-1 for absolute pressure values, but not for pressure tendency values (Fig. 6). |
|---|---|---|
| There are a number of regression equations (Equations 8, 9, 12) that appear to be tuned for Australia. Are users to rederive these regression equations for their domain of interest, or are they globally applicable? | Regression equations are based on Australian data, in line with the choice of case study. As the model is free and open source, it is possible for users to modify these regressions for their chosen domain. Supporting software demonstrates how the regression equations were derived, and can be used to determine those regressions for other regions | Added the following text to section 4: Regression models are used to control specific sub-components of the track model – Rmax, poci and landfall decay rate. These regression models are derived from observed data in the Australian region, but could equally be adapted to other regions. The code repository provides access to the analysis tools used to determine these regression models, and can be used to re-evaluate the regressions for other basins. The model is intended to be applied to regional basins, rather than a global domain, but the ability to adapt these regression models allows users to run in basins other than Australia.

 See also paragraph 1, page 2: "Where there are region-specific formulations in a component of the model, these can be readily adapted for different regions." |
| I don't understand the need for a decay rate model (Equation 10). Isn't the decay rate already included in the input best track (pmin) data? | Using the autoregressive model over land led to an abundance of storms that reintensified, and generated many more unrealistically long-lived tracks. Implementing the decay rate model forces the TCs to decay in line with observed events (but does allow for very occasional over-land intensification) | Added the following text to the beginning of section 4.5: Initial testing using only the autoregressive model for intensity after landfall resulted in unrealistically long-lived tracks after landfall. |

| | | |
|---|---|---|
| I think it would be useful to mention the option to additionally use local wind multiplication factors to better account for local terrain effects. | Noted | Added the following to section 5.2 (p. 7): For more localised wind speeds that can be used for detailed wind impact calculations (e.g. Krause and Arthur, 2018), local site conditions can be incorporated via an offline calculation that can incorporate local accelerations over topography and varying surface roughness conditions (Yang et al., 2014). |
| I read that it takes a few minutes to run a single scenario. Can some detail be added on the computational cost of running 1000 years? | The windfield for a single scenario takes about 10 minutes to run, depending on the overall extent of the track. Using multiprocessor systems, a hazard simulation, including track generation and wind fields across the Australian region (100-170E, 5-35S) for a 10,000 year catalogue requires around 3000 hours of CPU time. | Added the following to section 3: Simulation times are dependent on the extent of the domain, and the number of simulated years. For the domain used in this paper, the data processing and statistical analysis stages take around 15 minutes to complete on a modern desktop computer. The generation of tracks for a 10,000 year simulation takes around 5 to 6 CPU hours (2.6GHz clock speed), while the corresponding wind fields (a total of around 160,000 separate events for this simulation) take around 3,000 CPU hours. The determination of ARI wind speeds requires a similar amount of CPU time, but the majority is consumed in reading the required data from the wind field files. |
| I may have missed it, but I suggest including a statement that the model can also be used for single event scenario assessments? | The model can and is regularly used for single scenario simulation. At Geoscience Australia, we use the model to provide guidance on wind field zones for Emergency Management Authorities based on track information provided by the Bureau of Meteorology. We can provide the wind fields to EMs on average within 30 minutes of publication of track forecasts from the BoM. | See paragraph 2, page 2 |

| | | |
|---|---|---|
| I suggest adding that the model can be run with any input track data, not just historical best track data. This broadens the applications of the model to be used in conjunction with, for example, TC track data from global climate models to study climate variability and change effects on wind exceedances. | Noted - we are presently working on using other input sources to demonstrate this capability | Added the following to section 2: It is possible to use data sources other than observational best track archives as input to TCRM. For example, Siqueira et al. (2014) used tropical cyclone like vortices (TCLVs) extracted from global circulation models as a source of track data for evaluating TC wind hazard in the South West Pacific. After correcting the intensity distribution of the TCLV data, the resulting hazard assessment provided quantitative estimates of the projected change in TC wind hazard. |
| Section 4.5: It is stated that there are differences in the inland decay rates between the East and West coasts. But then a single decay rate model is used. Please justify this decision. | The driver of different decay rates in the east and west is topography. To minimise the data demands (especially with a view to global application), we did not include topography in the regression such that we do not have to source suitable topographic data for all potential basins. This is an area for future development. | Added the following to section 4.5: Initial testing using only the autoregressive model for intensity after landfall resulted in unrealistically long-lived tracks after landfall. Further, to explain the decision not to include topography in the model: In the interest of minimising the data demands (especially with a view to application in other basins), topography was not included in the regression such that we do not have to source suitable topographic data for all potential basins. However this is an area for future development. |
| Section 7.1: The somewhat poor performance of the model over Northwest Australia is explained by the lower genesis probabilities. How is it possible for the model to miss these local genesis patterns if it is sampling from the genesis probability surface? | Are track events being removed because of age or intensity issues? | Added Figure 6, plus the following text in section 4.1: The resulting genesis distribution of simulated events does not exactly match the historical distribution for a number of reasons (Fig. 6). Firstly, the stochastic sampling of the distribution for each simulated year will produce a different spatial pattern. In the case of simulating a large number of years, this would intuitively converge to the observed distribution. However, the |

| | | subsequent track behaviour determines if the track is retained – for example the simulated genesis density may be reduced in regions where tracks are excluded due to rapid weakening or exiting the domain. |
|---|---|---|
| Conclusion: The introduction mentions the high cost of riverine and coastal surge flooding. Can a brief discussion be added on whether a TCRM-like approach could be used for TC rainfall and/or flooding? | Noted. | Added the following to section 8 (Conclusion): The model provides information that can readily be used to guide other hazard assessments, such as wave climate modelling and coastal storm surge, and there is potential to include other perils such as rainfall through appropriate parametric models (Lonfat et al., 2007; Mudd et al., 2015). |
| Figure 5: Please add the units of the genesis probability. | Figure and caption updated | Figure and caption updated |
| Figure 13: What do the colors of the lines represent? | Caption updated | Caption updated |

| Reviewer 2 comments | Response comments | Changes |
|---|---|---|
| P2L30: "L is the bandwidth matrix." This sounds as if L varies through the domain. Is that right? If so, more detail is warranted, perhaps a sentence or two stating the range and typical values of L. | The bandwidth matrix is a 2x1 array that defines the bandwidth for x- and y-variation of the kernel function | Changed text at P4, paragraph 2 to: "L is a 2-by-2 bandwidth matrix determined automatically from the covariance of observed genesis points using a cross-validated maximum likelihood approach and is held constant over the entire simulation domain" |

| | | |
|---|---|---|
| Section 4.1. Genesis: Is there seasonality in the genesis? Or in the other model components? | Seasonality is considered in the genesis of events and the environmental pressure, but is not considered in other model components. To do so would detrimentally affect the sample sizes for the gridded statistical values applied in the track generation component. | Sentence added on P4, paragraph 3: "The annual cycle of genesis is included in determining the start time of TC events" |
| P6L39-P7L1: The assumption that Vtang » Vtran is often not good even for purely tropical systems, if they're not high intensity. | We have also noted this, and remain cautious of the resulting implications for ARI wind speeds at low intensity. | Added the following to section 5.2: The model assumes Vtang >> Vtran, which may be violated for low intensity storms (e.g. incipient TCs). The boundary layer model is modified to linearly reduce the effects of translation speed when Vtranslation > 0.2 Vtangential. The effects are also reduced to zero at distances greater than 2 Rmax, using an inverse square decay function. |
| P7L5: Additional recent relevant work on ET transition are documented in two papers by Bieli et al. (2019). The second paper, especially, addresses statistical modeling of ET transition. The references are Bieli et al, 2019, A global climatology of extratropical transition, Part I: characteristics across basins, J. Clim, 32, 3557-3582, and Bieli et al, 2019, A global climatology of extratropical transition, Part II: statistical performance of the cyclone phase space, J Clim, 32, 3583-35997) | Acknowledged. Most relevant is the pathways to ET, and the likelihood of each in different basins and the implications for parametric wind fields. | Added reference to Bieli et al, 2019 in section 5.2 (page 8, paragraph 3) |
| P8L28-31: I don't follow this. When I look at Fig 5 I do indeed see a local maximum in the 120-130E, 10S region of the genesis probability density function. What am I missing? | The genesis probability for simulations is not shown, only the observed genesis probability. | Added Figure 6, plus the following text in section 4.1: The resulting genesis distribution of simulated events does not exactly match the historical distribution for a number of reasons (Fig. 6). Firstly, the stochastic sampling of the distribution for each simulated year will produce a different spatial pattern. In the case of simulating a large number of years, this would intuitively converge to the observed distribution. However, the subsequent track behaviour determines if the track is retained – for example the simulated genesis density may be reduced in regions where tracks are excluded due to rapid weakening or exiting the domain. |

| | | |
|---|---|---|
| I'm surprised that the CP model seems to work well. One of the reasons that a track model like this works is that a TC's location strongly influences its propagation, due to the role of large-scale climatological steering winds. So, it's reasonable to estimate where a TC at location X goes next by analyzing where historical TCs near X have gone next, without any additional information. But this is less true for CP. Fig 11 shows essentially two zones, north a south of roughly 15S. North of that, the average CP tendency is negative, and south it's positive. Many TCs in the region spend the bulk of their time in the northern negative-tendency zone. For these TCs, what keeps their CPs from declining without limit? I realize there's a large stochastic component, but, without some cap, how is the TCs intensity bounded? For example, in their stochastic TC model, James and Mason (2005) employed an elastic cap at low CP to limit decline below the CP set by the local Maximum Potential Intensity. Perhaps here, given the proximity of landmasses in the region, there's rarely an opportunity for a TC's CP to decrease below plausible meteorological limits? | To minimise the volume of ancillary data that is needed to support the model initial development we aimed for a purely statistical approach. The lower limit for the pressure deficit is set to $\mu + 5*\sigma$ for the grid cell. However, we note that PI is potentially a more instructive limit, and we are presently working on enhancements that will consider this. | Added the following text to section 4.2: The maximum achievable central pressure of a TC is set to $[\![ \mu ]\!]\_p^i - 5\sigma\_p^i$, and is a purely statistical bound. However, we note that potential intensity (Holland, 1997) is potentially a more instructive limit, and we are presently working on enhancements that will consider this. p˙is preferred to absolute pressure deficit due to the lower lag-1 autocorrelation in the tendency values, making it more akin to a true Markov process than simulating absolute pressure deficit. Figure 7 shows the autocorrelation for both p˙and p for a selected grid cell in the Coral Sea. In this case, the lag-1 autocorrelation of p˙is 0.3, compared to that of p which is 0.79. Using absolute values leads to rapid and almost one-way variation (i.e. constant increase or decrease) in the intensity. There remains a strong autocorrelation beyond lag-1 for absolute pressure values, but not for pressure tendency values (Fig. 7). |
| Similarly, for the few TCs that form south of 15S, in the positive-tendency zone, what causes their CPs to decline on average (the storms to intensify), as opposed to just quickly attenuating to lysis? Is some kind of filter applied, to only accept storms that stochastically intensify beyond some threshold? | Storms that do not intensify beyond a pressure deficit of 5 hPa, or survive for more than 12 hours are discarded. | See paragraph 3 in section 4.1 (added text in italics): To allow for this in TCRM, weak lows are maintained if their central pressure deficit increases above 5 hPa within 12 hours of the initial time. This allows for initial formation over land *(or in areas of positive pressure tendency)*, as long as the incipient TC intensifies sufficiently (through the stochastic process described in the next section) – usually associated with a move over open water. |

| | | |
|---|---|---|
| Discussion about these points is warranted. In addition, it would be helpful to show some examples of over-ocean CP time series from stochastic simulations. It would also be helpful to have additional panels in Fig 19 that show landfall probabilities for TCs with CP below specified thresholds. (I assume the current version of Fig 19 applies to all TCs, regardless of CP.) Finally, I'd like to see the magnitude of the CP sigma, perhaps in a second panel of Fig 11, to compare to the mean tendencies. | | Added Figure 8, plus following text to section 4.2:
Figure 8 shows the time history of central pressure of a small sample of tracks that are generated from a single genesis point (155°E, 20°S) and the same initial central pressure (995 hPa). One storm weakens rapidly over the first 12 hours. The remaining storms take between 30 and 200 hours to attain maximum lifetime intensity. |
| Figs 1 and 19: It would help to have mileposts (e.g., towns) indicated on the landfall profile of Fig 19 and correspondingly on the map of Australia, perhaps the map of Fig 1. | Noted | Figure 1 updated with key locations and labels for gates |

[revised manuscript text omitted]

Commented [AC1]: Response to: I may have missed it, but I suggest including a statement that the model can also be used for single event scenario assessments?

Deleted: Further, the absolute accuracy of the input data is viewed as a source of uncertainty in the return period hazard values presented here. A thorough treatment of the accuracies is warranted (Harper et al., 2008), and the hazard values herein should be considered as only one view of the true wind hazard arising from TC events.

Commented [AC2]: Add discussion on uncertainty from short input record (and input data more generally), and how this should be considered in longer time scale contexts. Address stationary statistics issue (see below for more on other input sources)

[revised manuscript text omitted]

where $N$ is the number of genesis points, $d_i$ is the distance between genesis point $i$ and the point $(\lambda, \phi)$ (latitude and longitude). $L$ is a 2-by-2 bandwidth matrix determined automatically from the covariance of observed genesis points using a cross-validated maximum likelihood approach and is held constant over the entire simulation domain. The annual cycle of genesis is included in determining the start time of TC events.

This can result in simulation of genesis over land in the stochastic sampling step. In the Australian region, it is not unusual to observe the formation of precursor tropical lows over land. To allow for this in TCRM, weak lows are maintained if their central pressure deficit increases above 5 hPa within 12 hours of the initial time. This allows for initial formation over land (or in areas of positive pressure tendency), as long as the incipient TC intensifies sufficiently (through the stochastic process described in the next section) – usually associated with a move over open water.

The resulting genesis distribution of simulated events does not exactly match the historical distribution for a number of reasons (Fig. 6). Firstly, the stochastic sampling of the distribution for each simulated year will produce a different spatial pattern. In the case of simulating a large number of years, this would intuitively converge to the observed distribution. However, the subsequent track behaviour determines if the track is retained – for example the simulated genesis density may be reduced in regions where tracks are excluded due to rapid weakening or exiting the domain.

**4.2 Tracks**

Following determination of the initial location, intensity, translation speed and bearing of a TC event, the model applies an autoregressive process to step the TC forward in time. Equations 2 and 3 describe the translation speed of the TC located in grid cell $i$ at time $t$ (for the translation speed $v$):

$$v(t) = \mu_v^i + \sigma_v^i \chi^i(t), \tag{2}$$

$$\chi^i(t) = \alpha_v^i \chi^i(t-1) + \phi_v^i \varepsilon, \tag{3}$$

where $\mu_v^i$ is the observed mean translation speed $v$ in grid cell $i$, $\sigma_v^i$ is the observed standard deviation of $v$ and $\alpha_v^i$ is the observed lag-1 autocorrelation, and $\chi^i(t=0)=0$. $\phi_v^i$ controls the magnitude of the random variation $\varepsilon$ and is related to $\alpha_v^i$ through Eq. 4 (noting the change of use for the symbol $\phi$ from Eq. (1)):

$$\phi_v^{i\,2} = 1 - \alpha_v^{i\,2} \tag{4}$$

**Commented [AC5]:** Response to: There are a number of regression equations (Equations 8, 9, 12) that appear to be tuned for Australia. Are users to rederive these regression equations for their domain of interest, or are they globally applicable?

**Commented [AC6]:** Response to: "L is the bandwidth matrix." This sounds as if L varies through the domain. Is that right? If so, more detail is warranted, perhaps a sentence or two stating the range and typical values of L.

**Commented [AC7]:** Response to: Genesis: Is there seasonality in the genesis? Or in the other model components?

**Commented [AC8]:** Response to: Similarly, for the few TCs that form south of 15S, in the positive-tendency zone, what causes their CPs to decline on average (the storms to intensify), as opposed to just quickly attenuating to lysis? Is some kind of filter applied, to only accept storms that stochastically intensify beyond some threshold?

[revised manuscript text omitted]

**Commented [AC9]:** Response to: For these TCs, what keeps their CPs from declining without limit? I realize there's a large stochastic component, but, without some cap, how is the TCs intensity bounded? For example, in their stochastic TC model, James and Mason (2005) employed an elastic cap at low CP to limit decline below the CP set by the local Maximum Potential Intensity.

**Commented [AC10]:** Response to : Please explain why the time rate-of-change of central pressure is used rather than the absolute value of central pressure?

are derived using randomly selected values of $\Delta p$ and $\lambda$. The model slightly over predicts $R_{max}$ at low intensity ($20 < \Delta p < 40$ hPa), but otherwise provides an excellent match to the observations.

**4.4 Pressure of outermost closed isobar**

The central pressure deficit $\Delta p$ used to quantify the intensity of synthetic TCs is the difference between the central pressure and the pressure of the outermost closed isobar $p_{oci}$. We initially considered the daily long-term mean sea level pressure at the location of the TC ($p_{ltm}$) as a proxy for $p_{oci}$. However, there are substantial and systematic differences between the two (Fig. 10). Using $p_{ltm}$ will lead to synthetic TCs generating sufficient wind speeds to remain defined as TCs at higher central pressure values than observed. To define $p_{oci}$ for the synthetic TCs, we modify $p_{ltm}$ based on the central pressure, latitude and day of year, plus a random innovation:

$$p_{oci} = 2324.2 - 0.65399 p_{ltm} - 1.398 p_c + 0.000740 p_c^2 + 0.00445 \lambda^2 - 1.434 \cos\left(2\pi d_{year}/365\right) + \varepsilon, \qquad (9)$$

where $p_{ltm}$ is the daily long term mean sea level pressure at the location of the TC, $p_c$ is the central pressure, $\lambda$ the latitude, $d_{year}$ the day of year. $\varepsilon$ is a random innovation sampled from a normal distribution with $\mu = 0$ and $\sigma = 2.572$. The coefficients were determined using ordinary least squares fitting to the parameters, using observed values of $p_{oci}$ from 2002-2016 JTWC records (n=1833).

Modelled values of $p_{oci}$ qualitatively match the observed values (Fig. 11), with $l^2$ norm values all less than 0.4. Closer inspection however reveals subtle differences. When plotted against $p_{ltm}$ (Fig. 11a), the maximum density of modelled values of $p_{oci}$ is skewed to lower values (approximately 3 hPa lower). For $p_c$ versus $p_{oci}$ (Fig. 11b), the comparison is much closer, with the peaks in the PDF for both modelled and observed $p_{oci}$ coinciding near weak (high $p_c$) and $p_{oci}$ near 1006 hPa. Comparison by latitude (Fig. 11c) is very good, with the peak of the PDF of modelled values overlaying the observed peak. The PDF of modelled $p_{oci}$ against day of year (Fig. 11d) is also very close to the observed distribution.

**4.5 Landfall**

Initial testing using only the autoregressive model for intensity after landfall resulted in unrealistically long-lived tracks after landfall. Instead, the filling rate of TCs after landfall is modelled in the same manner as Vickery (2005), where the central pressure deficit $\Delta p$ decreases as an exponential function of time over land $t$, the central pressure deficit at landfall $\Delta p_0$ and the translation speed at landfall, $v_0$:

$$\Delta p(t) = \Delta p_0 \exp(-\alpha t) \qquad (10)$$

where

$$\alpha = \alpha_0 + \alpha_1 \Delta p_0 + \alpha_2 v_0 \qquad (11).$$

To determine an optimum value for the parameters $\alpha_0$, $\alpha_1$ and $\alpha_2$, the decay behaviour of 174 landfalling TCs recorded in the IBTrACS dataset was analysed (Fig. 12). $\Delta p_0$ is the last observation of central pressure deficit prior to landfall, and all observations of $\Delta p$ after landfall are normalised by this value. Differences in the decay rate of TCs can be identified between those making landfall on the northwest Australian coastline and the eastern coastline (Fig. 13). We hypothesise that this is due to the presence of the Great Dividing Range along the eastern coast of Australia, with elevations exceeding 1400 metres in places (e.g. Mt Bartle Frere and Bellenden Ker). Further, the mean central pressure of landfalling cyclones in eastern Australia is higher than those along the western Australian coastline (Fig. 14). In the interest of minimising the data demands (especially with a view to application in other basins), topography was not included in the regression such that we do not have to source suitable topographic data for all potential basins. However this is an area for future development.

Commented [AC11]: Response to: I don't understand the need for a decay rate model (Equation 10). Isn't the decay rate already included in the input best track (pmin) data?

Commented [AC12]: Response to: It is stated that there are differences in the inland decay rates between the East and West coasts. But then a single decay rate model is used. Please justify this decision.

An exponential decay function was fitted to the normalised pressure deficit $\Delta p / \Delta p_0$ for the 174 landfalling TCs (Fig. 15). In general, $\Delta p$ follows the expected exponential decay form with $\alpha$ defined as:

$$\alpha = 0.03515 + 0.000435\Delta p_0 + 0.002865v_0 + \varepsilon(\mu, \sigma) \tag{12}$$

where $\varepsilon$ is a random variate sampled from a lognormal distribution with $\mu$=0.6953 and $\sigma$=0.0471, and held fixed for each event. Coefficients were fitted using non-linear least squares optimisation. This gives a decay rate parameter that is influenced by central pressure at landfall and replicates the observed decay rates well (Fig. 16). The effect of the landfall decay model can also be seen in several of the storms in Fig. 8. Storms that move back over open water revert back to the stochastic intensity model, with some storms showing reintensification. For example, track 0 makes landfall after about 220 hours, weakens, but reverts back to the stochastic intensity model near 235 hours, before a second landfall at 242 hours.

**4.6 Lysis**

Lysis of a synthetic TC occurs when $\Delta p$ falls below an arbitrary threshold, set to be 5 hPa, either due to the decline in intensity following landfall, or through the autoregressive process described above. TCs are also terminated on exiting the track domain.

**5 Tropical cyclone wind field model**

Parametric wind fields are calculated for each event in the synthetic catalogue to enable a high spatial resolution understanding of the ARI wind speeds. The additional benefit of this calculation is that users can select individual synthetic events from the catalogue and obtain a wind field for use in scenario simulations.

> **Commented [AC13]:** Response to: It's not clear to me the value of running the wind field model vs. simply running more synthetic years to build up enough tracks in each analysis grid cell. For example, what is the difference in the 500-year wind speed based on 100 tracks in each grid cell (no windfield module) and 100 wind field values in each grid cell (associated with tracks within and just outside each grid box)?

**5.1 Radial wind profile**

The wind field around each TC is calculated at high spatial resolution (up to 0.01°) to ensure the peak wind speeds near the eye are accurately captured. TCRM first uses a radial profile to estimate the gradient level wind associated with the vortex. To allow users to explore the range of variability in ARI wind speeds associated with different radial profiles, we have implemented a number of profiles in TCRM. These include the Holland (1980), Schloemer (1954), Willoughby and Rahn (2004), Powell et al. (2005), Jelesnianski (1966), the McConochie et al. (2004) double exponential profile and a Rankine vortex profile. The Willoughby, Schloemer and Powell et al. profiles are all variants of the Holland profile – the difference being the definition of the peakedness or β parameter. While more complex radial profiles are available in the literature, we have chosen to implement simpler models that rely only on readily available best-track parameters (e.g. central pressure, latitude). For this verification study, the Powell et al. (2005) profile was used, with β defined as:

[revised manuscript text omitted]

**Commented [AC14]:** Response to: The assumption that Vtang » Vtran is often not good even for purely tropical systems, if they're not high intensity.

**Commented [AC15]:** Response to: I think it would be useful to mention the option to additionally use local wind multiplication factors to better account for local terrain effects.

**Commented [AC16]:** Most relevant is the pathways to ET, and the likelihood of each in different basins and the implications for parametric wind fields.

[revised manuscript text omitted]

> **Commented [AC17]:** Response to: It would also be helpful to have additional panels in Fig 19 that show landfall probabilities for TCs with CP below specified thresholds. (I assume the current version of Fig 19 applies to all TCs, regardless of CP.)

[revised manuscript text omitted]